# Numerical stabilization methods for level-set-based ice front migration

Gong Cheng[1], Mathieu Morlighem[1], and G. Hilmar Gudmundsson[2]

[1]Department of Earth Sciences, Dartmouth College, Hanover, NH 03755, USA
[2]Department of Geography and Environmental Sciences, Northumbria University, Newcastle upon Tyne, UK

**Correspondence:** Gong Cheng (gong.cheng@dartmouth.edu)

**Abstract.** Numerical modeling of ice sheet dynamics is a critical tool for projecting future sea-level rise. Among all the processes responsible for the loss of mass of the ice sheets, enhanced ice discharge triggered by the retreat of marine terminating glaciers is one of the key drivers. Numerical models of ice sheet flow are therefore required to include ice front migration in order to reproduce today's mass loss and be able to predict their future. However, the discontinuous nature of calving poses a significant numerical challenge for accurately capturing the motion of the ice front. In this study, we explore different stabilization techniques combined with varying reinitialization strategies to enhance the numerical stability and accuracy of solving the level-set function, which tracks the position of the ice front. Through rigorous testing on an idealized domain with a semicircular and a straight-line ice front, including scenarios with diverse front velocities, we assess the performance of these techniques. The findings contribute to advancing our ability to model ice sheet dynamics, specifically calving processes, and provide valuable insights into the most effective strategies for simulating and tracking the motion of the ice front.

## 1 Introduction

Ice sheet numerical modeling is the best tool to make future sea-level rise projections (e.g., Seroussi et al., 2020; Goelzer et al., 2020; IPCC, 2021). One key process that significantly contributes to mass loss is the retreat of marine terminating glaciers (Mouginot et al., 2019; Choi et al., 2021; Pattyn and Morlighem, 2020). For example, in Greenland, the increased ice discharge is mainly driven by the retreat of glacier fronts (King et al., 2020), which is a direct consequence of calving and undercutting at the ice front (Wood et al., 2021; Mouginot et al., 2019), possibly intensified by increased runoff and ocean temperatures (Black and Joughin, 2023). As Greenland has very few ice shelves, ice front retreat predominantly comprises small yet frequent calving events (Black and Joughin, 2023; Cheng et al., 2021). Future projections emphasize that ice front retreat will continue to be a primary driver of Greenland's mass loss by 2100 (Choi et al., 2021). Incorporating moving boundaries into numerical ice sheet models is a vital step in advancing our understanding of ice loss mechanisms and improving the accuracy of future sea-level rise projections (Crawford et al., 2021; Bondzio et al., 2017; Cheng et al., 2022).

Ice sheets are commonly modeled as incompressible fluids governed by conservation laws (e.g., Greve and Blatter, 2009), with empirical calving laws to predict calving rates at the ice front (Pollard et al., 2015; Morlighem et al., 2016). These calving laws are parameterizations developed based on physical principles and observations, which offer computationally efficient and relatively straightforward expressions for calving rates (Benn and Astrom, 2018; Choi et al., 2018). In these parameterizations, the boundary of the model, which is generally the ice front, needs to be adjusted dynamically during the transient simulation. The way ice front migration is typically handled is through a level-set function, which is a signed distance function defined over the entire computational domain with the zero level-set contour representing the ice front position (e.g., Bondzio et al., 2016; Morlighem et al., 2016). The motion of the level-set function is determined by solving an advection equation, where the difference between the ice velocity and the calving (and melting) rate at the zero-contour governs the evolution (Morlighem et al., 2016).

However, solving numerically the level-set function is challenging, especially when using the finite element method (FEM), as it can lead to instabilities due to the unbounded gradient of the solution (Larson and Bengzon, 2013). To address this issue, stabilization techniques are employed to enforce the boundedness of the solution. Additionally, the transient solution of the level-set function may not always maintain its signed distance property due to inhomogeneities in the velocity field and the accumulation of numerical errors over time, particularly through the diffusion introduced by the stabilization method. Therefore, reinitialization is generally necessary during transient simulations to restore the signed distance function property. However, as highlighted in Henri et al. (2022), reinitialization may introduce an artificial subsequent displacement of the zero level-set contour. Therefore, the selection of the reinitialization interval is critical for obtaining an accurate solution of the signed distance function, which remains inherently dependent on the specific application.

In this paper, we aim to investigate and compare various stabilization techniques in combination with different choices of reinitialization intervals, implemented in the Ice-sheet and Sea-level System Model v4.23 (ISSM, Larour et al., 2012; ISSM Team, 2023) and Úa 2019b (Gudmundsson et al., 2019; Gudmundsson, 2020). We present different stabilization and reinitialization procedures, and apply them all in ISSM to solve the level-set equation on an idealized domain featuring a semicircular ice front shape (and a straight-line ice front shape case in the appendix) representative of typical Greenland outlet glaciers. To evaluate the effectiveness of the stabilization techniques and reinitialization strategies, we perform several tests on three different spatially varying rates of ice front migration, encompassing both low and high-speed scenarios. By exploring these approaches, we seek to investigate which combination leads to the best stability and accuracy of simulating the level-set function and effectively tracks the motion of the ice front in ice sheet models.

## 2   Methods

The level-set function $\phi(\boldsymbol{x}, t)$ is a scalar field defined on a two-dimensional domain $\Omega$ with zero contours implicitly representing the ice front position at every given time $t$. Conventionally, the level-set function is set to be negative in the ice-covered region and positive in the ice-free region (Morlighem et al., 2016), in order for the gradient of the level-set to be normal outward pointing to the ice front. The absolute value of the level-set is the closest distance from $\boldsymbol{x}$ to the ice front contour $\phi = 0$. Given

an initial condition $\phi(\boldsymbol{x}, 0) = \phi_0$, the evolution of the level-set function $\phi(\boldsymbol{x}, t)$ is governed by the advection equation

$$\frac{\partial \phi}{\partial t} + \boldsymbol{v}_f \cdot \nabla \phi = 0, \quad \boldsymbol{x} \in \Omega, \, t \in [0, T] \tag{1}$$

where $\boldsymbol{v}_f$ is the front velocity of the level-set, which is the difference between the ice velocity, $\boldsymbol{v}$, and the calving rate of $c$, which is generally oriented perpendicular to the ice front:

$$\boldsymbol{v}_f = \boldsymbol{v} - c \, \mathbf{n}, \tag{2}$$

where $\mathbf{n}$ is the outward unit normal vector of the level-set (Bondzio et al., 2016; Morlighem et al., 2016).

In order to solve Eq. (1) with the FEM, we introduce a Hilbert space $\mathcal{H}^1(\Omega)$ and define the variational form as: find $\phi \in \mathcal{H}^1(\Omega)$ such that for all the test function $\psi \in \mathcal{H}^1(\Omega)$ the equation

$$\int_\Omega \left( \frac{\partial \phi}{\partial t} \psi + (\boldsymbol{v}_f \cdot \nabla \phi) \psi \right) \, \mathrm{d}\Omega = 0, \tag{3}$$

is satisfied. After replacing the space $\mathcal{H}^1(\Omega)$ by a continuous piecewise linear space $\Phi_h$, the solution of Eq. (3) is then the numerical solution of Eq. (1). However, it is well known that Eq. (3) gives spurious oscillatory solutions without stabilization (Larson and Bengzon, 2013; dos Santos et al., 2021).

## 2.1 Stabilization

We consider four stabilization schemes in this paper. The first three methods are classical methods only to stabilize Eq. (3), namely, artificial diffusion (MacAyeal, 1989, AD), streamline upwinding (Eriksson, 1996, SU), and, Streamline Upwinding Petrov-Galerkin(Brooks and Hughes, 1982, SUPG). The last one is a modification of the SUPG stabilization, where an additional *forward-and-backward* (Li et al., 2005, FAB) diffusion term is added to the SUPG scheme.

Among them, the simplest way to stabilize an advection equation is to add an additional diffusion term in the variational form Eq. (3) such that

$$\int_\Omega \left( \frac{\partial \phi}{\partial t} \psi + (\boldsymbol{v}_f \cdot \nabla \phi) \psi + \nabla \phi \cdot \boldsymbol{\kappa} \nabla \psi \right) \, \mathrm{d}\Omega = 0, \tag{4}$$

where, in two dimensions, the coefficient of the artificial diffusion term is a scalar

$$\kappa = \frac{1}{2} \sqrt{h_x^2 v_x^2 + h_y^2 v_y^2}, \tag{5}$$

where $h_x$ and $h_y$ are the characteristic mesh sizes in $x$ and $y$ directions, $v_x$ and $v_y$ are the $x$ and $y$ components of the front velocity $\boldsymbol{v}_f$.

The streamline upwinding stabilization follows the same variational form as the artificial diffusion in Eq. (4), but with a modified coefficient derived from Eq. (5). Specifically, this modification ensures the addition of diffusion solely along the direction of the velocity vector $\boldsymbol{v}_f$ by using

$$\boldsymbol{\kappa} = \frac{h}{2\|\boldsymbol{v}_f\|} \boldsymbol{v}_f \otimes \boldsymbol{v}_f, \tag{6}$$

where $h = \sqrt{h_x^2 + h_y^2}$ and $\otimes$ is the Kronecker product. Due to the large dissipation introduced by these two stabilization methods, they are extremely stable but only have first-order accuracy (dos Santos et al., 2021).

A more accurate stabilization method is the Streamline upwind Petrov–Galerkin (SUPG, Brooks and Hughes, 1982), which modifies the test function to be $\hat{\psi} = \psi + \mu \boldsymbol{v}_f \cdot \nabla \psi$ in the variational form in Eq. (3) such that

$$\int_\Omega \left( \frac{\partial \phi}{\partial t} + \boldsymbol{v}_f \cdot \nabla \phi \right) (\psi + \mu \boldsymbol{v}_f \cdot \nabla \psi) \; \mathrm{d}\Omega = 0, \tag{7}$$

where $\mu = \frac{h}{2\|\boldsymbol{v}_f\|}$ is a mesh dependent coefficient (dos Santos et al., 2021).

The FAB diffusion was first introduced in Úa (Gudmundsson et al., 2019; Gudmundsson, 2020). We follow the same formulation and implement it in ISSM. The FAB term added to the variational form in Eq. (7) is derived from the potential

$$\mathcal{P} = \frac{1}{pq} \int_\Omega (\|\nabla \phi\|^q - 1)^p \, d\Omega \tag{8}$$

for which the directional derivative is

$$D_{\delta\phi} \mathcal{P} = \int_\Omega (\|\nabla \phi\|^q - 1)^{p-1} \|\nabla \phi\|^{q-2} \nabla \phi \cdot \nabla \delta \phi \, d\Omega \tag{9}$$

This results in the addition of a non-linear diffusion term to the level-set equation, with a diffusion coefficient

$$\kappa = \mu (\|\nabla \phi\|^q - 1)^{p-1} \|\nabla \phi\|^{q-2}, \tag{10}$$

which is bounded for $\|\nabla \phi\| \to 0$, provided $q \geq 2$. For even values of $p$, the diffusion term defined by Eq. (10) can be both negative and positive and is an example of a FAB diffusion. Note that the minimum of the potential $\mathcal{P}$ in Eq. (8) is found for $\|\nabla \phi\| = 1$, i.e. when $\phi$ is a distance function. This approach therefore encourages the level set to remain a signed distance function and relaxes the need to reinitialize the level set.

## 2.2 Reinitialization

The formulation of the advection equation Eq. (1) describes the evolution of the level-set function, however, it does not guarantee that the level-set function is always a signed *distance* function due to the inhomogeneity of the front velocity. Indeed, as $\boldsymbol{v}_f$ is generally higher at the ice front than the far field, the gradient of the level-set function close to the zero contours tends to decrease during the transient simulation.

To maintain the gradient of the level-set function, a common practice is to reset the level-set by calculating the signed distance every $n_R$ time steps. This is often called 'reinitialization' (Bondzio et al., 2016; Morlighem et al., 2016), and the reinitialization interval $n_R$ is the number of time steps between two consecutive reinitializations. One method of reinitialization involves solving an Eikonal equation (Sussman et al., 1994; Sethian, 1996):

$$\|\nabla \phi\| = 1, \tag{11}$$

generally expressed as a time-dependent problem for which we seek a steady state solution:

$$\frac{\partial \phi}{\partial t} + \text{sign}(\phi)(\|\nabla\phi\| - 1) = 0. \tag{12}$$

However, this approach (12) contains control parameters, and it is not clear what the optimal value of these parameters should be in practical application (Gross and Reusken, 2011). Moreover, the Eikonal equation constitutes a nonlinear hyperbolic partial differential equation (PDE), posing challenges in achieving accurate discretization. An alternative is to use the Fast Marching Method (Sethian, 1996; Toure and Soulaimani, 2016). This method offers a general framework capable of handling various scenarios.

Here, we use a straightforward geometric reinitialization algorithm, which is similar to the one described in Toure and Soulaimani (2016). At any point in time, the zero contour of the level set is represented by a set of segments if it is discretized using linear Lagrange elements. At the reinitialization step, we create a loop over all elements and generate this set of segments, with one segment per element containing a change in the sign of the level-set function. This set of segments is then shared across all model partitions through a Message Passing Interface, in order to recompute a signed distance. Subsequently, at each vertex of the mesh, we compute the distances to these segments and keep the minimum distance as the new magnitude of the level-set at that vertex, while preserving the original sign. When this reinitialization algorithm is applied, it is expected to yield exact results in terms of signed distance. Hence, we do not expect that the proposed FAB diffusion algorithms would outperform this method in terms of accuracy. However, as we show later in the numerical experiments, numerical errors highly depend on the reinitialization frequency. Here, we investigate different reinitialization intervals combined with the four stabilization methods described in Section 2.1.

## 2.3 Error Quantification

In order to quantify the difference between two ice front positions represented by the level-set functions $\phi_1$ and $\phi_2$, we introduce a misfit metric $d(\phi_1, \phi_2)$ such that

$$d(\phi_1, \phi_2) = \frac{\text{sgn}(\phi_1) - \text{sgn}(\phi_2)}{2}, \tag{13}$$

where

$$\text{sgn}(\phi) = \begin{cases} -1, & \phi < 0, \\ 0, & \phi = 0, \\ 1, & \phi > 0, \end{cases} \tag{14}$$

converts a level-set function to a sign function with $-1$ on the ice-covered side of the zero contour and $1$ on the ice-free side of the contour. Therefore, if $\phi_1$ is ahead of $\phi_2$ in terms of the ice front positions (more advance), the misfit area in $d(\phi_1, \phi_2)$ will be negative.

We integrate the absolute misfit over the whole domain, $\Omega$, and get a metric

$$\mathcal{J}(\phi_1, \phi_2) = \int_\Omega |d(\phi_1, \phi_2)| \, d\Omega = \frac{1}{2} \int_\Omega |\text{sgn}(\phi_1) - \text{sgn}(\phi_2)| \, d\Omega, \tag{15}$$

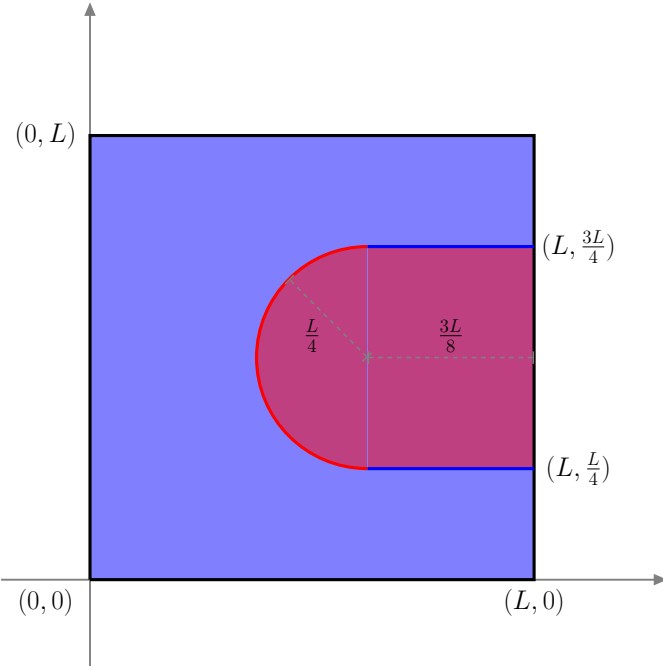

**Figure 1.** The domain of the semicircle-shaped ice front. The light blue area indicates the ice-covered region and the light red area is ice-free. The values close to the dashed grey lines are their lengths.

which is actually the absolute misfit area between the two level-set functions.

## 3   Numerical experiments

We investigate the influence of the four stabilization methods described in Section 2.1 combined with different choices of reinitialization interval (Section 2.2). We consider here a semicircle-shaped initial ice front as shown in Figure 1, where the

ice-covered region is in light blue, and the ice-free region is in light red. We apply analytical spatially and temporally varying velocity fields to mimic typical ice flow.

   We run all the simulations on a two-dimensional square domain $\Omega(x, y) = [0, L] \times [0, L]$, with $L = 20$ km as the size of the domain. We create an unstructured triangular mesh on $\Omega$ with the element size of 100 m. In Figure 1, the calving front is represented by a semicircle (red) centered at $(c_x, c_y) = (\frac{5L}{8}, \frac{L}{2})$ with a radius of $r = \frac{L}{4}$, and the sidewalls of the fjord are in blue

and connect the semicircle to the right boundary of the domain. By construction, the width of the fjord is 10 km. The initial zero level-set is the red ice front together with the blue sidewalls, which has a closed form as $\{(x, y) | (x - c_x)^2 + (y - c_y)^2 = r^2, x \leq c_x\} \bigcup \{(x, y) | x \in [c_x, L], y = c_y + r\} \bigcup \{(x, y) | x \in [c_x, L], y = c_y - r\}$.

   We apply three distinct velocity fields to control the migration of the ice front. For simplicity, we assume that there is no ice flux across the side walls of the fjord so that the velocity field only contains a horizontal component as $\boldsymbol{v}_f = (v_x, 0)^T$. The

$x$-component of the velocity fields are given in Table 1. They represent zeroth (uniform), first (triangle), and second (parabola) order polynomials shape of the velocities.

| Shape | Formula |
| --- | --- |
| Uniform | $v_x(x,y,t) = v(t)$ |
| Triangle | $v_x(x,y,t) = v(t)\left(1 - \left\lvert\frac{y}{c_y} - 1\right\rvert\right)$ |
| Parabola | $v_x(x,y,t) = v(t)\left(1 - \left(\frac{y}{c_y} - 1\right)^2\right)$ |

**Table 1.** The three shapes of front velocity at the ice front.

Temporal variations are introduced by flipping the sign of $v(t)$ (as in Table 1) every half year to mimic the typical annual cycle of the advance and retreat of an ice front such that

$$v(t) = \begin{cases} v_0, & t \in [nT, (n+\frac{1}{2})T), \\ -v_0, & t \in [(n+\frac{1}{2})T, (n+1)T), \end{cases} \tag{16}$$

where $T = 1$ year, $n = 0, 1, 2, 3, ..., N$ and $v_0$ is a velocity constant. We examine two scenarios with high ($v_0 = 5000$ m/a) and low ($v_0 = 1000$ m/a) velocity constants, respectively. All the simulations are run for $N = 50$ periods (or years), with a constant time step at $\Delta t = 0.005$ year to satisfy the CFL condition for both of the high and low-velocity scenarios. We reinitialize the zero level-set contour with the interval $n_R = 1, 10, 100, 200$, which corresponds to a reinitialization every 2 days, two-thirds of a month, half a year, and one year. We also set a control run with no reinitialization ($n_R = \infty$) throughout the whole simulation
period.

By applying the velocity for $\frac{T}{2}$ in one direction, then flipping the sign of $v_x$ for another $\frac{T}{2}$, the ice front is expected to return to its initial position $\phi_0$ after every period $T$. Furthermore, the analytical solution at any given time $t + nT$ should be identical to the solution at time $t$. Therefore, we use the numerical solution at $t \in [0, T)$ as the exact solution, and calculate the numerical error at $t + nT$ according to Eq. (15), with $\phi_1 = \phi(\boldsymbol{x}, t + nT)$ and $\phi_2 = \phi(\boldsymbol{x}, t)$.

**4   Results**

The misfit between the numerical and the exact solution under a uniform velocity field at the low-velocity setting ($v_0 = 1000$ m/a) after 1.5, 2 and 50 periods (or years) are shown in Figure 2 with $n_R = 1$, and in Figure 3 with $n_R = 100$ for the four stabilization methods considered in this paper. The misfit at every time point is calculated according to Eq. (13), where the area with negative values (blue in the figures) indicates the ice front from the numerical solution is downstream (i.e. further
advanced) of the exact solution. The errors of all the cases in Figure 2 and 3 are almost evenly distributed along the ice front,

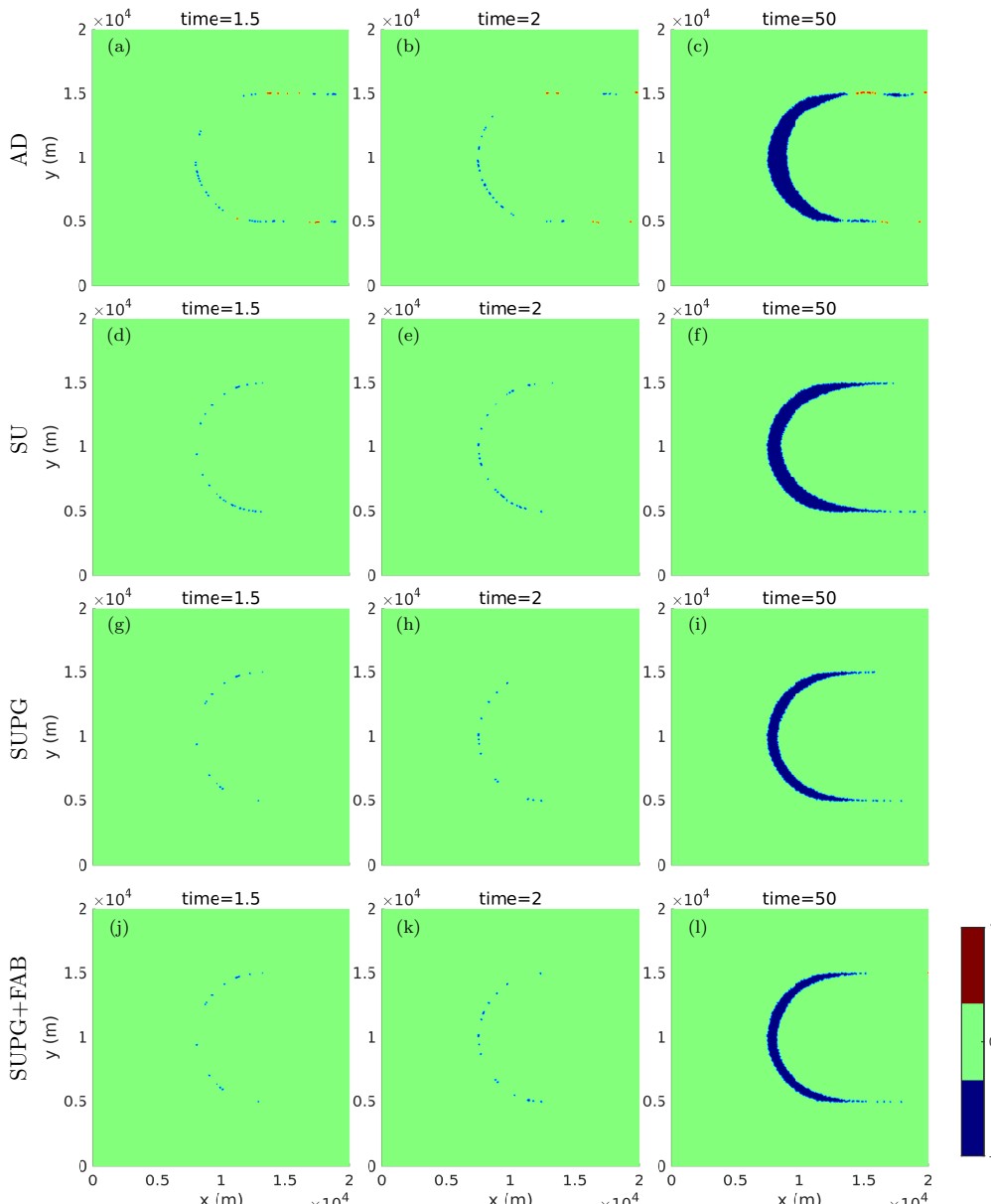

**Figure 2.** Misfit $d(\phi_1, \phi_2)$ of the numerical solution at time $t$ (as $\phi_1$) and its exact solution (as $\phi_2$) at the reinitialization interval $n_R = 1$, and $v_0 = 1000$ m/a, with (a-c) AD, (d-f) SU, (g-i) SUPG, and (j-l) SUPG+FAB stabilizations.

and the total misfit grows as time increases. Indeed, all the errors are first-order in time, as we show the time series of the errors in the Appendix B for different stabilizations, reinitializations, and velocity constants. Figures 2 and 3 also indicate that using $n_R = 100$ gives more accurate results compared to reinitializing every time step ($n_R = 1$).

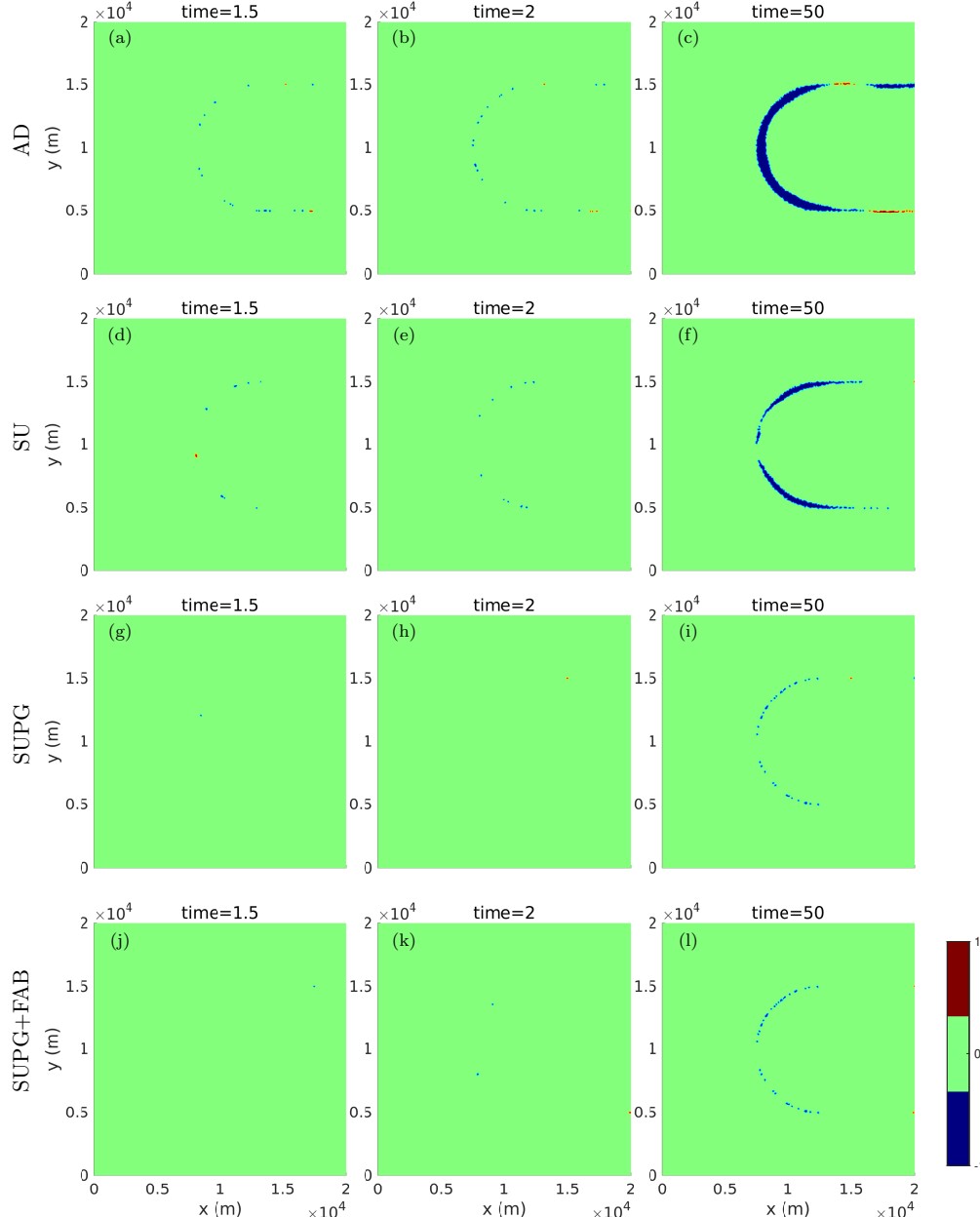

**Figure 3.** Misfit $d(\phi_1, \phi_2)$ of the numerical solution at time $t$ (as $\phi_1$) and its exact solution (as $\phi_2$) at the reinitialization interval $n_R = 100$, and $v_0 = 1000$ m/a, with (a-c) AD, (d-f) SU, (g-i) SUPG, and (j-l) SUPG+FAB stabilizations.

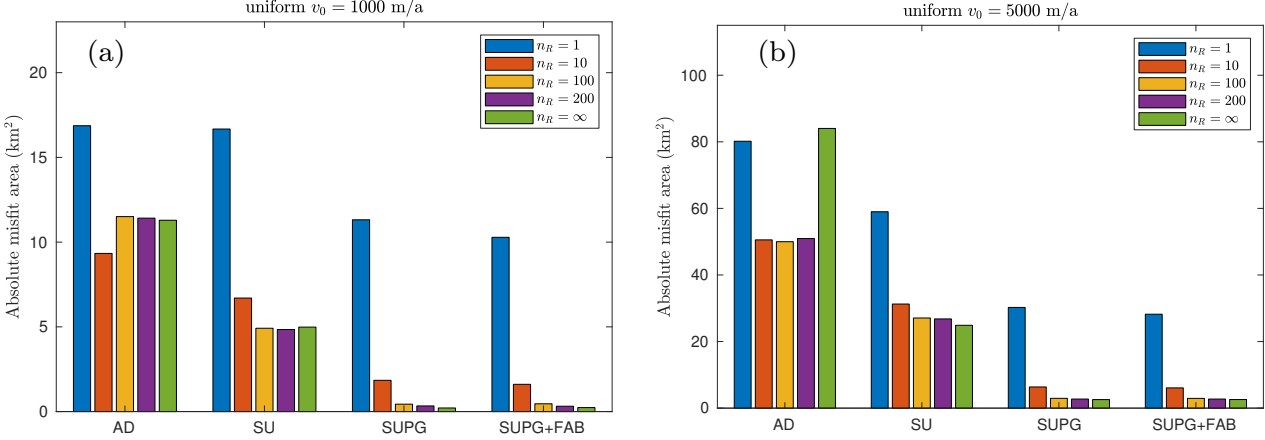

**Figure 4.** Total absolute misfit area at $T = 50$ for semicircle front with uniform velocity (a) $v_0 = 1000$ m/a and (b) $v_0 = 5000$ m/a. The $y$-axis in (b) is scaled by a factor of five for visualization purposes

To facilitate a better comparison of the different stabilization, reinitialization, and velocity constant choices, we show the total absolute misfit in Figure 4, which is calculated according to Eq. (15) at the final time step $t = 50$ years for the uniform velocity field. The numerical errors tend to decrease as the reinitialization interval $n_R$ increases. Specifically, in Figure 4 (a), the four largest errors occur when the level-set function is reinitialized at every time step ($n_R = 1$), resulting in errors of 16.87 km$^2$ in AD, 16.67 km$^2$ in SU, 11.32 km$^2$ in SUPG, and 10.28 km$^2$ in SUPG+FAB. The spatial distributions of the errors are shown in Figure 2 (c), (f), (i), and (l). Given that the width of the fjord is 10 km, these errors correspond to an average offset of the ice front of 1 to 2 km along the flow direction. After $n_R > 10$, the numerical errors remain almost constant, comparable to the ones of $n_R = \infty$, for all the stabilization methods employed. We find a similar pattern in the high-velocity ($v_0 = 5000$ m/a) cases in Figure 4 (b), where most of the numerical errors are approximately five times larger than those in the low-velocity ($v_0 = 1000$ m/a) cases in Figure 4 (a). However, the high-velocity cases are less sensitivity to $n_R$ than the low-velocity cases. For instance, reinitializing every time step does not introduce exceptionally large errors as we found in the low-velocity cases. Indeed, the largest numerical error (88.62 km$^2$) among all the experiments is achieved by the AD stabilization without reinitialization.

Although all four stabilization methods tend to overestimate the advance of the ice front, the choice of stabilization method has a significant impact on the misfit area, and SUPG+FAB exhibits the lowest numerical errors. In the low-velocity scenario, e.g. Figure 4 (a), with $n_R = 100$, the final misfit for SUPG+FAB is 0.46 km$^2$, whereas the errors for AD, SU, and SUPG are 11.51 km$^2$, 4.92 km$^2$, and 0.44 km$^2$, respectively. The spatial distributions of these errors are shown in Figure 3 (c), (f), (i), and (l), where the misfit achieved by SUPG is equivalent to an offset of the ice front by approximately 46 m, which is even less than half of the mesh size. Similarly, in the high-velocity scenario, the errors are scaled by the front velocity in all the choices of stabilizations with $n_R > 1$. For instance, in Figure 4 (b), with $n_R = 100$, the errors are 49.99 km$^2$ in AD, 27.07 km$^2$ in SU, 2.94 km$^2$ in SUPG, and 2.92 km$^2$ in SUPG+FAB.

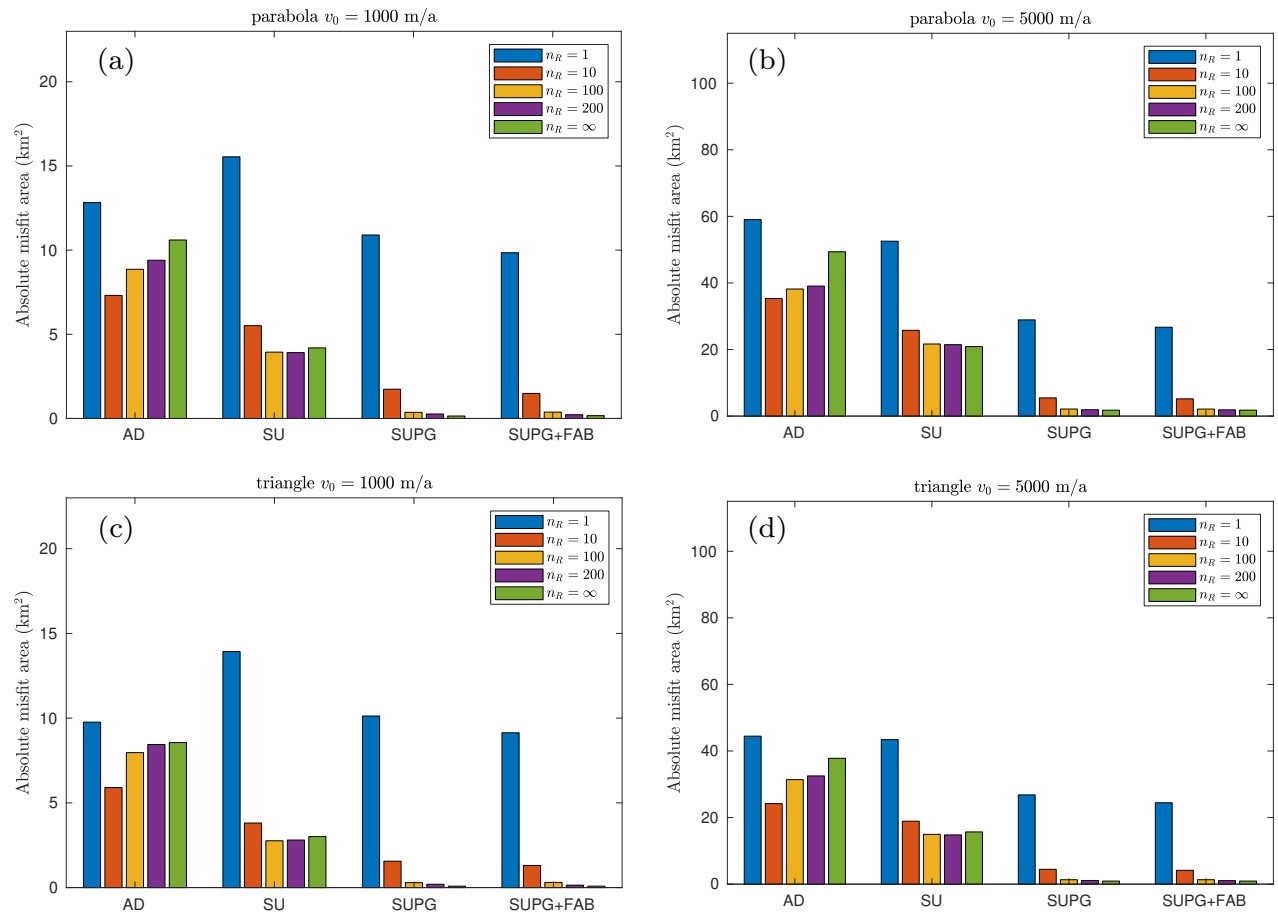

**Figure 5.** The total absolute misfit area at $T = 50$ with (a, b) parabola and (c, d) triangle shape velocity profiles. The left column has the velocity constant $v_0 = 1000$ m/a, and the right column is at $v_0 = 5000$ m/a.

We present the numerical errors at the final time step for the parabolic and triangular shape of velocity in Figure 5 for both low and high-velocity constants. Apparently, the shape of the velocity profile has a limited impact on the numerical errors. Nevertheless, the triangular velocity cases yield the smallest errors, while the parabolic velocity cases yield larger errors, but still smaller than the uniform velocity field scenario depicted in Figure 4.

## 5 Discussion

### 5.1 Reinitialization interval

From a finite-element method point of view, the reinitialization procedure is an $\mathcal{L}^2$ projection of the zero level-set contour onto the mesh (Larson and Bengzon, 2013). It can be shown that the numerical errors of the projection are proportional to

the mesh sizes (shown in Figure C1), and they accumulate as the number of reinitializations increases (i.e., as $n_R$ decreases). Furthermore, these errors are not only introduced during the projection but also transported and amplified by the governing

equation Eq. (1) throughout the transient simulation. In the case of frequent reinitializations, such as $n_R = 1$, the dominant source of numerical error is the $\mathcal{L}^2$ projection, particularly evident when the front velocity is low ($v_0 = 1000$ m/a), as depicted in Figures 4 and 5. However, in the high-velocity scenario, the projection error becomes less significant compared to the numerical errors resulting from discretization and stabilization techniques, which then become the primary sources of error.

As $n_R$ increases, the numerical error decreases until no reinitialization is performed ($n_R = \infty$). However, in the absence of

215 reinitialization, additional errors emerge due to the distortion of the gradient of the level-set function. The worst-case scenario observed in this study is the high uniform velocity case with AD at $n_R = \infty$ in Figure 4 (b), where the zero-contour of the final level-set solution is nearly halfway into the fjord, resulting in a total misfit of $84.02$ km$^2$. This instance emphasizes the necessity to reinitialize the level-set when solving level-set functions in transient simulations. It is worth noting that the numerical errors are not significantly affected by the interval of reinitialization as long as $n_R$ is sufficiently larger than 1. Consequently, for the

220 remainder of this paper, the focus will be on discussing the cases with $n_R = 10, 100$, and $200$, while disregarding those with $n_R = 1$ and $n_R = \infty$

As discussed above, the FAB penalizes deviations from the Eikonal equation, ensuring $\|\nabla\phi\| = 1$ when solving the level-set (Hartmann et al., 2010). The reinitialization interval is crucial in determining how often the level-set needs to be reset using the geometric reinitialization algorithm described in section 2.2. A naïve approach would be to reinitialize the level set after

225 each steps of solving the advection equation in order to maintain its signed-distance property. However, in practice, frequent reinitialization introduces interface displacements due to numerical errors, resulting in artificial mass gain or loss, which may also alter the geometrical characteristics of the interface, with potential implications for topological changes (Hartmann et al., 2010; Gibou et al., 2018; Henri et al., 2022).

## 5.2 Stabilization method

The numerical errors in AD and SU are 5 to 20 times greater than those using SUPG and FAB as long as $n_R$ exceeds 1. The main source of the numerical errors in AD and SU is the diffusion term $\nabla\phi \cdot \boldsymbol{\kappa}\nabla\psi$ added in the advection equation Eq. (4), which smears out the oscillations in the numerical solution and disperses the solution. The coefficient $\boldsymbol{\kappa}$ controls the magnitude and direction of the additional diffusion.

In the AD case, the coefficient $\boldsymbol{\kappa}$ is a scalar, which applies the diffusion to all directions with the same magnitude. In contrast,

$\boldsymbol{\kappa}$ contains an outer product of the front velocity in SU, which only adds diffusion along the flow direction of $\boldsymbol{v}_f$. Therefore, the errors in SU are less dispersive than those in AD. Notably, the coefficients $\boldsymbol{\kappa}$ in AD and SU are also controlled by the mesh size, such that the additional diffusion term vanishes as the mesh size becomes zero. In numerical ice sheet modeling, the mesh size is generally limited by data accuracy and computational capacity. Therefore, the weak solution of the stabilized equation Eq. (4) does not necessarily satisfy the variational formulation Eq. (3), and the corresponding errors are proportional to the

mesh size (Larson and Bengzon, 2013).

On the other hand, the SUPG stabilizes the advection equation by adding an additional term in the test function as in Eq. (7), whose solution satisfies the weak form Eq. (3) almost everywhere, except for the position where the test functions equal to 0. In this sense, the numerical error is expected to be much smaller than the other two stabilization methods. We therefore recommend using SUPG for the stabilization technique, together with a reinitialization interval greater than 10.

## 5.3 Front velocity

We anticipate the numerical errors to be scaled by the velocity magnitude when solving the advection equation using the finite element method (Biswas et al., 1994), but not influenced by the shape of the calving front. As we construct the velocities in Table 1, for instance, with $v_0 = 1000$ m/a, the mean frontal velocity during the advance phase $t \in [nT, (n + \frac{1}{2})T]$ is 1000 m/a for the uniform shape, 916.7 m/a for the parabola and 750.0 m/a for the triangular shape. The corresponding numerical errors, at $n_R = 100$ with SUPG stabilization, are 0.44 km$^2$, 0.36 km$^2$, and 0.29 km$^2$, respectively. Furthermore, as shown in Figures 4 and 5, this relationship is found in almost all the reinitialization intervals $n_R > 1$, all stabilization techniques, and both the low and high-velocity scenarios considered in this study.

Note that, while we do not model calving explicitly in this paper, the definition of the frontal velocity in Eq. (2) relies on $\boldsymbol{v}_f$, which implicitly incorporates the effects of calving or calving rate. It is important to distinguish that the velocity of the front ($\boldsymbol{v}_f$) is not the same as the calving velocity. The frontal velocity, $\boldsymbol{v}_f$, is a sum of the ice speed (which is not necessarily normal to the ice front) and the calving rate $c$, which is generally defined along the normal $\boldsymbol{n}$. Therefore, the ice front velocity is not necessarily orthogonal to the front in practice.

This study primarily focused on comparing different stabilization and reinitialization strategies for solving the level-set equation, assuming that $\boldsymbol{v}_f$ is known (i.e., both ice velocity and calving rates are known). The main purpose of this study is to demonstrate that even with a simple prescribed frontal velocity, stabilization and reinitialization can have a significant impact depending on the choices made. Incorporating a realistic calving term may not necessarily provide additional insights into our study, as it is already accounted for through $\boldsymbol{v}_f$ in the level-set equation. Moreover, introducing a calving law would preclude the availability of analytical solutions, complicating the interpretability of our results. As a future continuation of this study, in the CalvingMIP project (https://github.com/JRowanJordan/CalvingMIP/wiki), the ice sheet modeling comminity is testing more realistic calving velocities on more complex geometries, including constant and time-dependent calving rates.

## 5.4 Different front shapes

In Appendix A, we show the results of another shape of the ice front, which is a straight line with side walls orthogonal to the front. The final errors of the straight front cases with different stabilization methods, reinitialization intervals, and velocity shapes are more or less the same as those with the semicircle front. However, the spatial distribution of the numerical error differs significantly between the two shapes. To further investigate the source of the numerical errors, we show the animations of the evolution of misfits in the supplementary material. In the straight front cases, the misfit is initiated at the two corners, where the ice front meets the side wall of the fjord, and then propagates to the center. In contrast, the semicircle case generates numerical errors that do not initiate from single sources, but grow along the entire ice front. The main reason for

these differences is that the finite element method approximates the level-set function by projecting it onto a piecewise linear functional space. As a result, the sharp corners and the curved level-set contours are the places where most of the numerical errors occur. On average, these approximation errors are proportional to the mesh size, whereas the shape of the ice front actually has a negligible influence on the numerical errors.

## 6    Conclusions

We studied multiple stabilization methods implemented in ISSM and Úa for solving a level-set equation on an idealized geometry with a reinitialization interval that varies from once every time step up to no reinitialization. We found that SUPG and SUPG+FAB are considerably more accurate than the other two methods (AD and SU), for all choices of reinitialization interval, regardless of the front velocity and ice front shape. Using other stabilization methods results in more than ten times larger errors in ice front positions. An optimal choice for the reinitialization interval is $n_R > 10$, corresponding to a time period exceeding 2.5 weeks in our experiments. Excessively frequent reinitialization can introduce additional numerical errors surpassing those from other sources. By identifying the most effective stabilization techniques and reinitialization intervals, we can improve the reliability and robustness of simulations, enabling more accurate predictions of ice sheet behavior and its influence on future sea-level rise.

*Code availability.* ISSM Version 4.23 is open source and available at https://doi.org/10.5281/zenodo.7850841 (ISSM Team, 2023). Úa (v2019b) is open source and available at https://doi.org/10.5281/zenodo.3706624 (Gudmundsson, 2020). The code and data analyses used in this manuscript are available at https://doi.org/10.5281/zenodo.10454657.

*Video supplement.* The supplement video of the evolution of the misfit are available at https://doi.org/10.5281/zenodo.10454554.

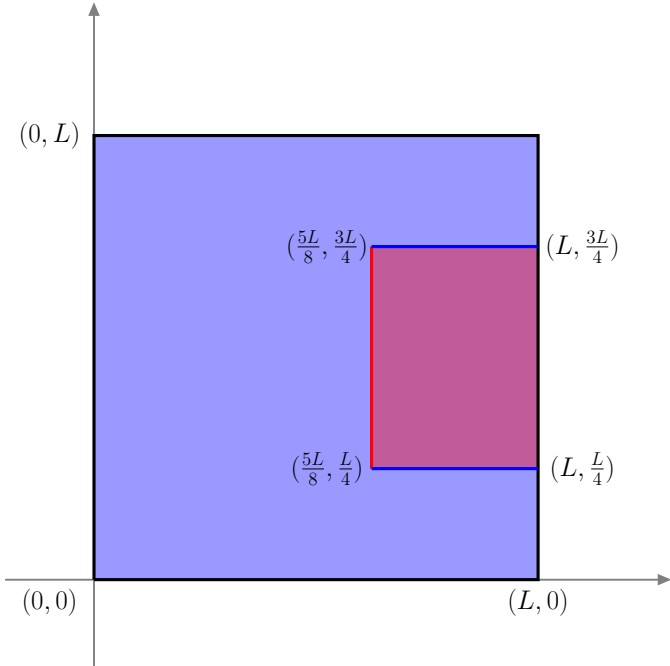

**Figure A1.** The domain of the straight ice front with the coordinates of the vertices.

## Appendix A: A straight ice front case

We introduce an alternative ice front shape, represented as a straight line, as depicted in Figure A1. Similar to Figure 1, the ice-covered region is denoted in light blue, while the ice-free region is in light red. The red line signifies the ice front, and the blue lines represent the side walls of the fjord, with a width of 10 km and a length of 20 km. The same set of experiments outlined in Section 3 is conducted, and the total misfit at the final time step is presented in Figure A2.

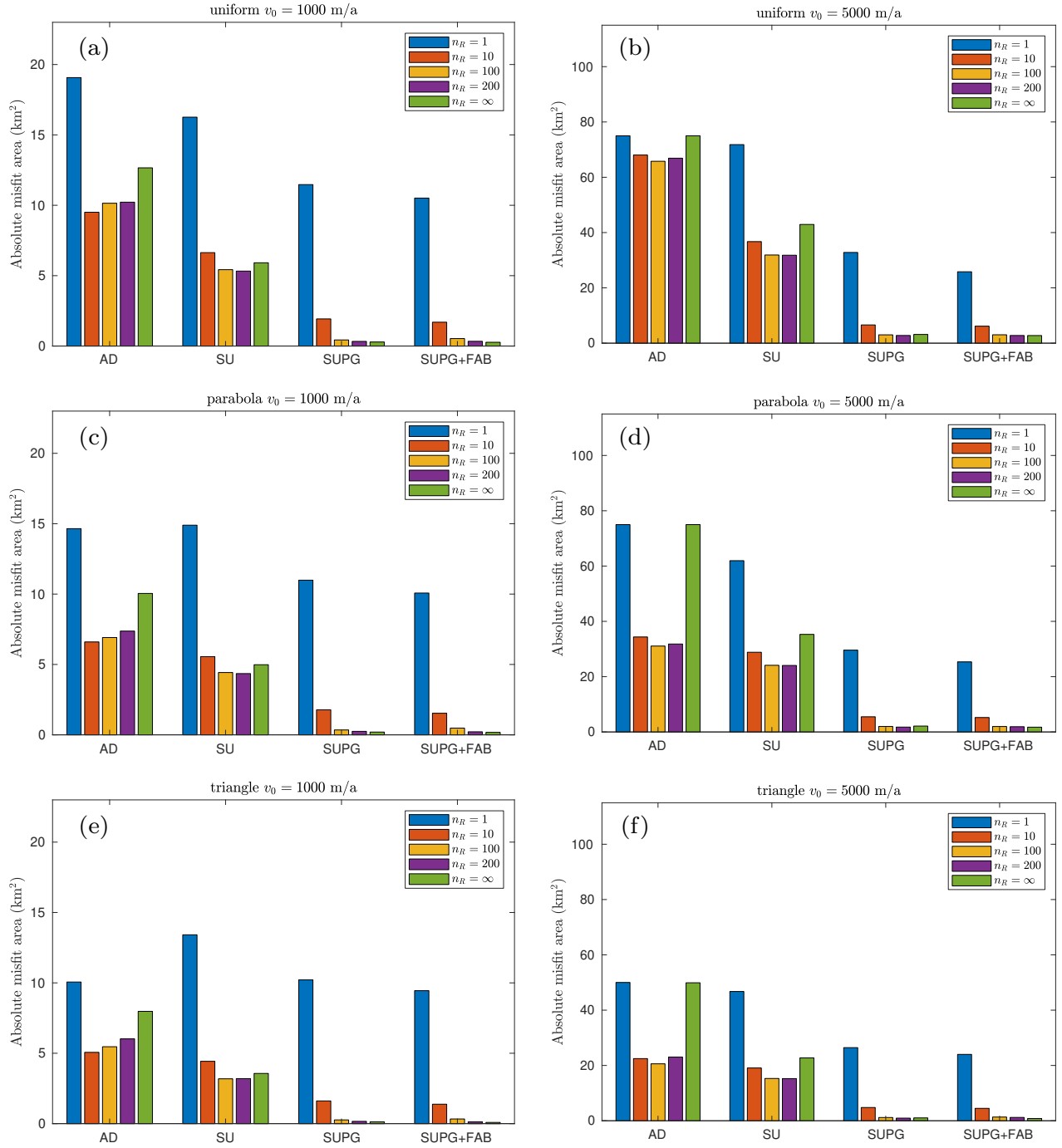

**Figure A2.** The total absolute misfit area at $T = 50$ with (a, b) uniform, (c, d) parabola, and (e, f) triangle shape velocity profiles for a straight ice front. The left column is in the low-velocity scenario with $v_0 = 1000$ m/a, and the right column is at $v_0 = 5000$ m/a.

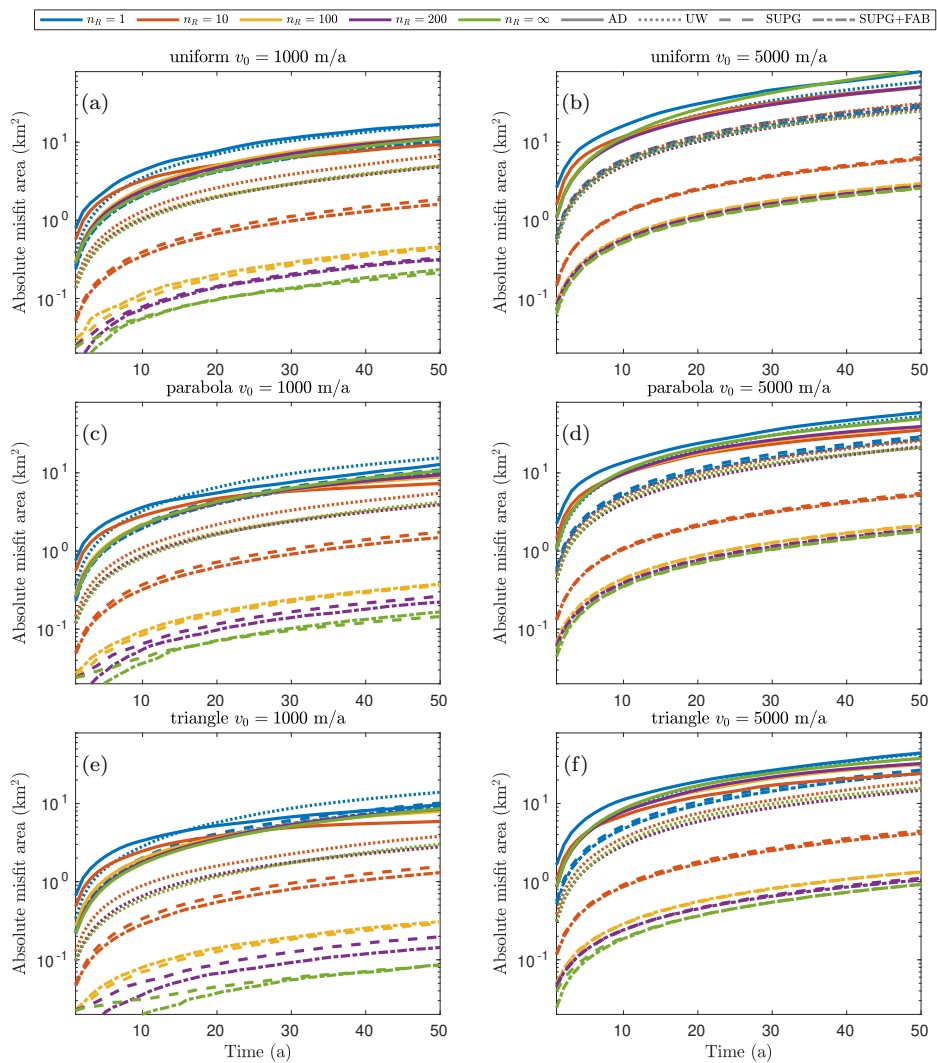

**Figure B1.** The evolution of the total absolute misfit area during the transient simulations with (a, b) uniform, (c, d) parabola, and (e, f) triangle shape velocity profiles for a semi-circle shape ice front. The left column is in the low-velocity scenario with $v_0 = 1000$ m/a, and the right column is at $v_0 = 5000$ m/a.

## Appendix B: Errors during the transient simulation

The numerical errors exhibit a linear scaling in time, as illustrated in Figures B1 and B2 across nearly all cases. As expected, the slopes are dictated by the velocity $v_0$. Consequently, for the sake of simplicity in comparison, we exclusively consider the numerical errors at the final time step $T = 50$ in the main text of this manuscript.

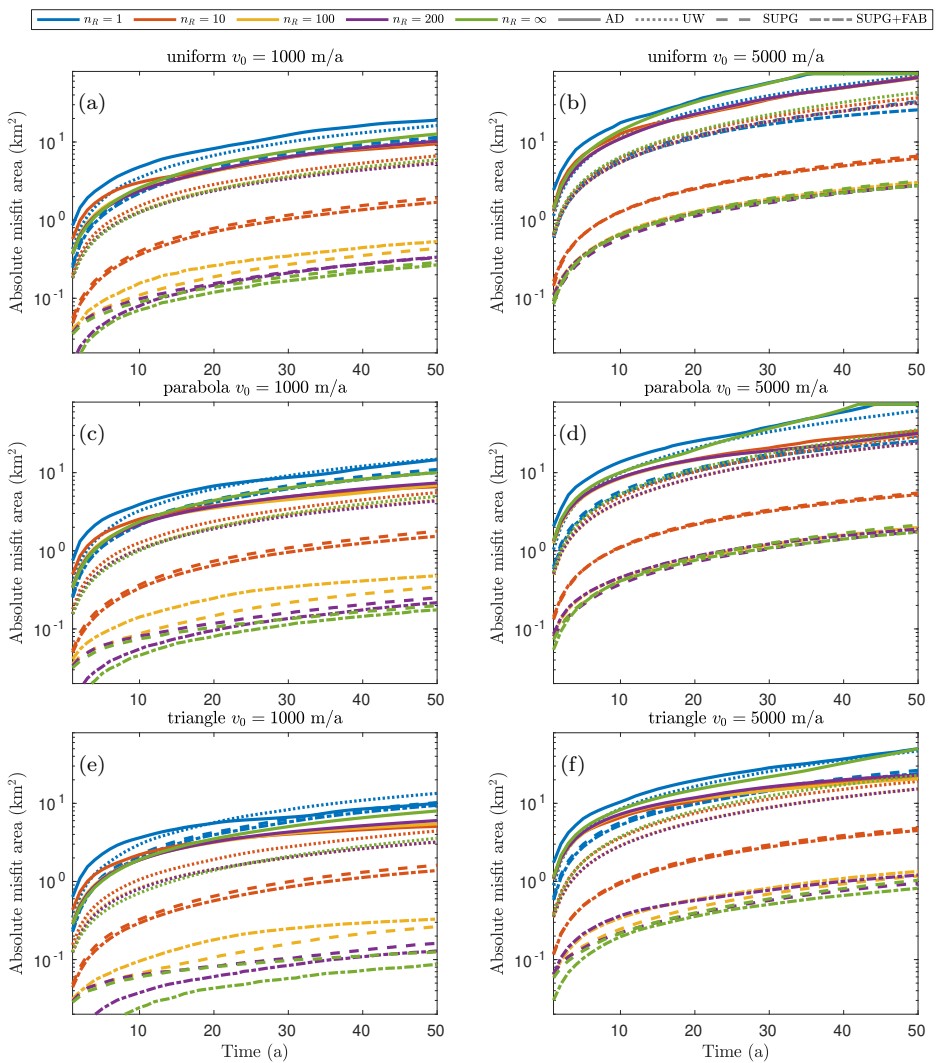

**Figure B2.** The evolution of the total absolute misfit area during the transient simulations with (a, b) uniform, (c, d) parabola, and (e, f) triangle shape velocity profiles for a straight ice front. The left column is in the low-velocity scenario with $v_0 = 1000$ m/a, and the right column is at $v_0 = 5000$ m/a.

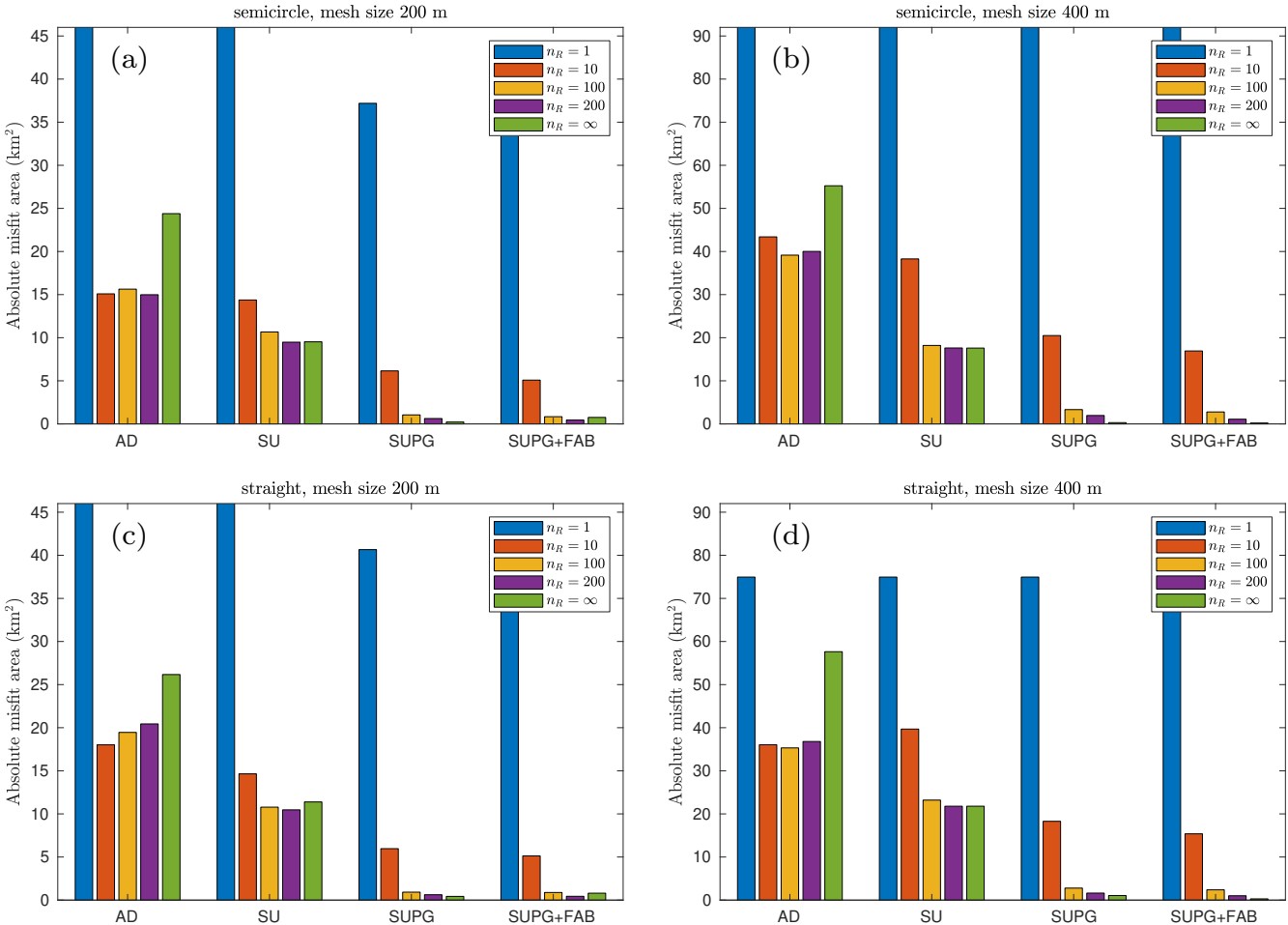

**Figure C1.** The total absolute misfit area at $T = 50$ with uniform velocity profile at $v_0 = 1000$ m/a for (a, b) semi-circle shape ice front, and (c, d) straight ice front. The left column is at 200 m mesh resolution, and the right column is at 400 m mesh resolution.

## Appendix C: Mesh resolution

We also conducted this study using different mesh resolutions, namely 200 m and 400 m, and the corresponding numerical errors are depicted in Figure C1. To facilitate comparison, we scaled the y-axis by a factor of 2 and 4 for the two mesh resolutions, respectively. As anticipated, the comparison with results in Figures 4 and A2 reveals a linear scaling of numerical errors with the mesh size. Notably, in Figure C1 (d), $n_R = 1$ for all four stabilization methods reaches the maximum possible error, equivalent to the area of the fjord in the straight ice front case, i.e., 75 km$^2$.

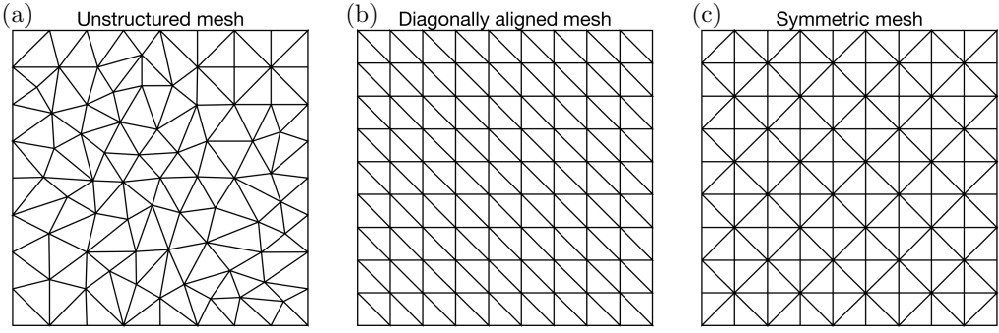

**Figure D1.** Diagrams of the employed meshes: (a) Unstructured mesh, (b) Diagonally aligned triangular mesh, and (c) Symmetric triangular mesh.

## Appendix D: Numerical errors influenced by the mesh structure

We investigate the effects of structured meshes on the level-set solutions by conducting two additional sets of experiments with different mesh configurations. One experiment employs a diagonally aligned triangular mesh, while the other relies on a symmetric triangular mesh. The mesh illustrations are presented in Figure D1, where (a) represents the unstructured mesh used in this study. Figure D1 (b) depicts a diagonally aligned mesh extending from the top left to the bottom right, while Figure D1 (c) shows a symmetric triangular mesh.

Figure D2 shows the misfit between the numerical and exact solutions for the diagonally aligned triangular mesh with a mesh resolution of 100 m, under a uniform velocity field with $v_0 = 1000$ m/a, observed after 1.5, 2, and 50 periods. Conversely, Figure D3 presents the results of the same experiment but on a symmetric triangular mesh. There is a clear asymmetry in the results shown in Figure D2 when using the diagonally aligned triangular mesh, but not for the unstructured mesh (e.g., Figure 2) or the symmetric triangular mesh in Figure D3.

To further examine the diagonally aligned triangular mesh, we refine the mesh resolution to 50 m and 25 m. Figure D4 shows the misfit between the numerical and exact solutions observed after 1.5, 2, and 50 periods, with AD stabilization, at $n_R = 1$ under a uniform velocity field with $v_0 = 1000$ m/a. Although numerical errors decrease with mesh refinement, the asymmetric error patterns persist even at a 25 m resolution, which is nearly the finest mesh resolution used in real world applications. This experiment highlights the importance of the mesh structure, particularly when geometric reinitialization is performed, as it may significantly depend on the organization and orientation of the elements of the mesh.

## Appendix E: Additional experiment with a more realistic frontal velocity

We introduce an additional numerical experiment aimed at further improving the realism of the frontal velocity representation. In this experiment, we modify the frontal velocity in Eq. (16) to $v(t) = v_0 \sin(2\pi t)$, simulating seasonal variations rather than abrupt transitions. This adjustment seeks to emulate the dynamic movement of ice fronts influenced by seasonal changes.

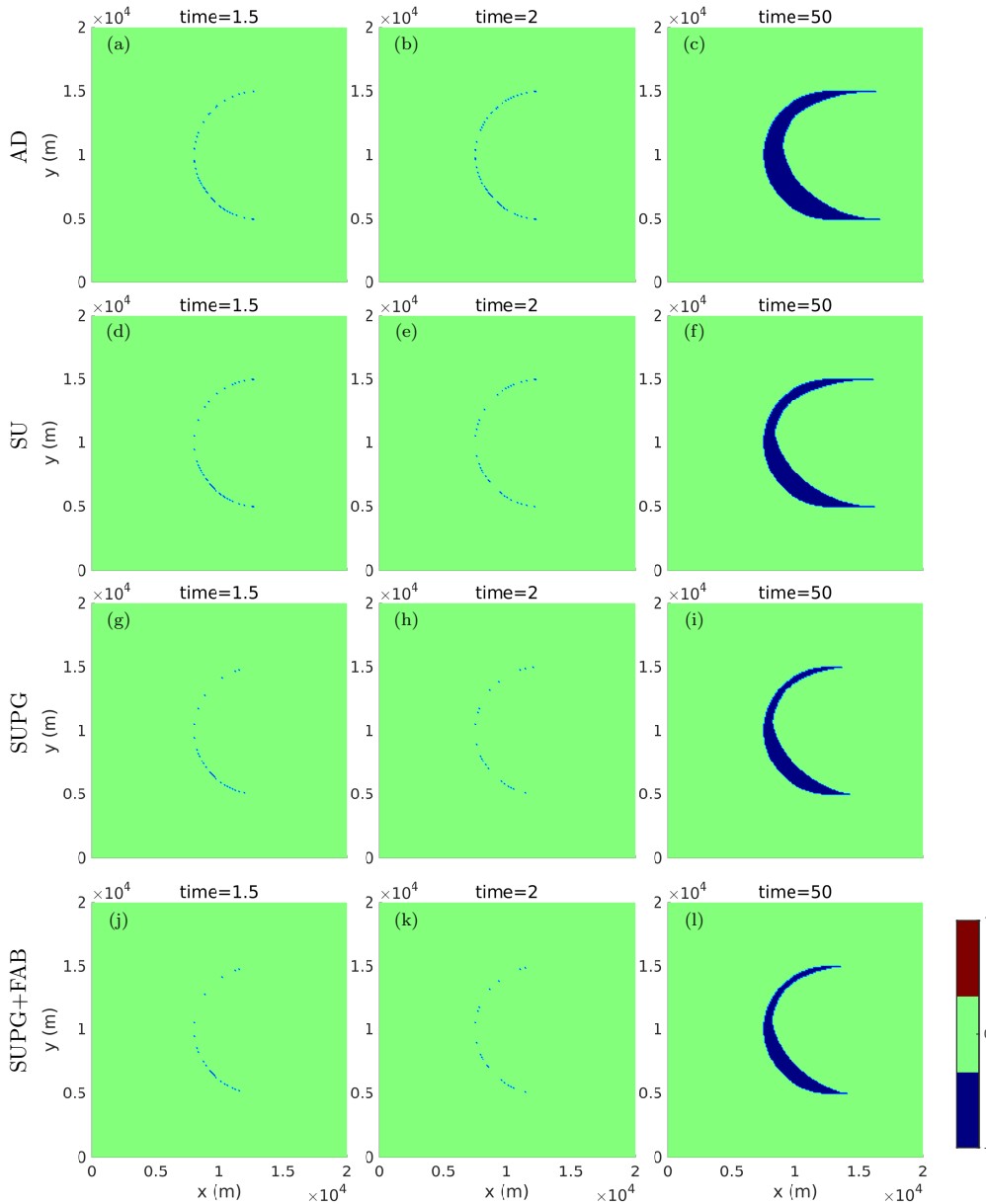

**Figure D2.** Misfit $d(\phi_1, \phi_2)$ of the numerical solution at time $t$ (as $\phi_1$) and its exact solution (as $\phi_2$) on a diagonally aligned triangular mesh at the reinitialization interval $n_R = 1$, and $v_0 = 1000$ m/a, with (a-c) AD, (d-f) SU, (g-i) SUPG, and (j-l) SUPG+FAB stabilizations.

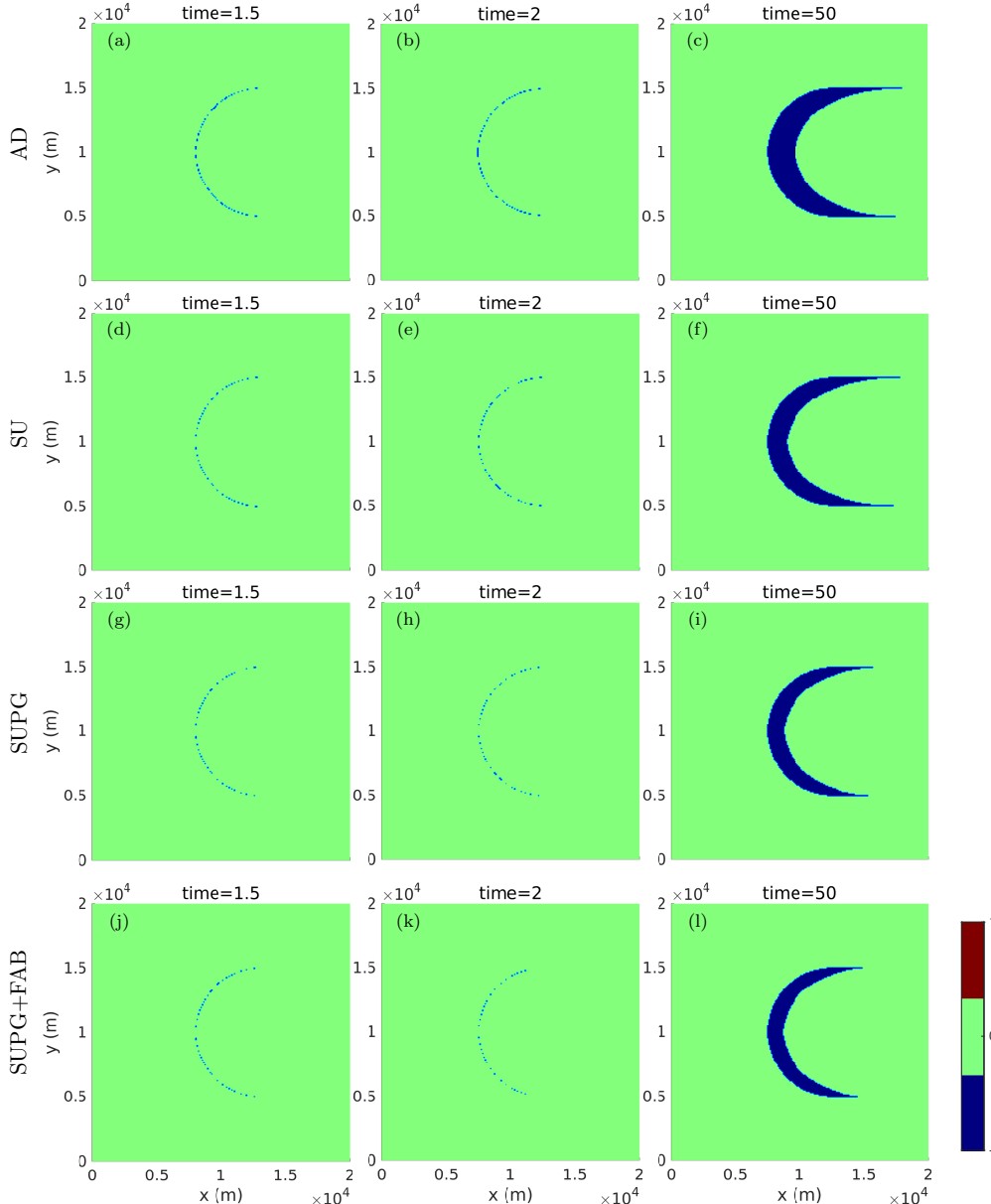

**Figure D3.** Misfit $d(\phi_1, \phi_2)$ of the numerical solution at time $t$ (as $\phi_1$) and its exact solution (as $\phi_2$) on a symmetric triangular mesh at the reinitialization interval $n_R = 1$, and $v_0 = 1000$ m/a, with (a-c) AD, (d-f) SU, (g-i) SUPG, and (j-l) SUPG+FAB stabilizations.

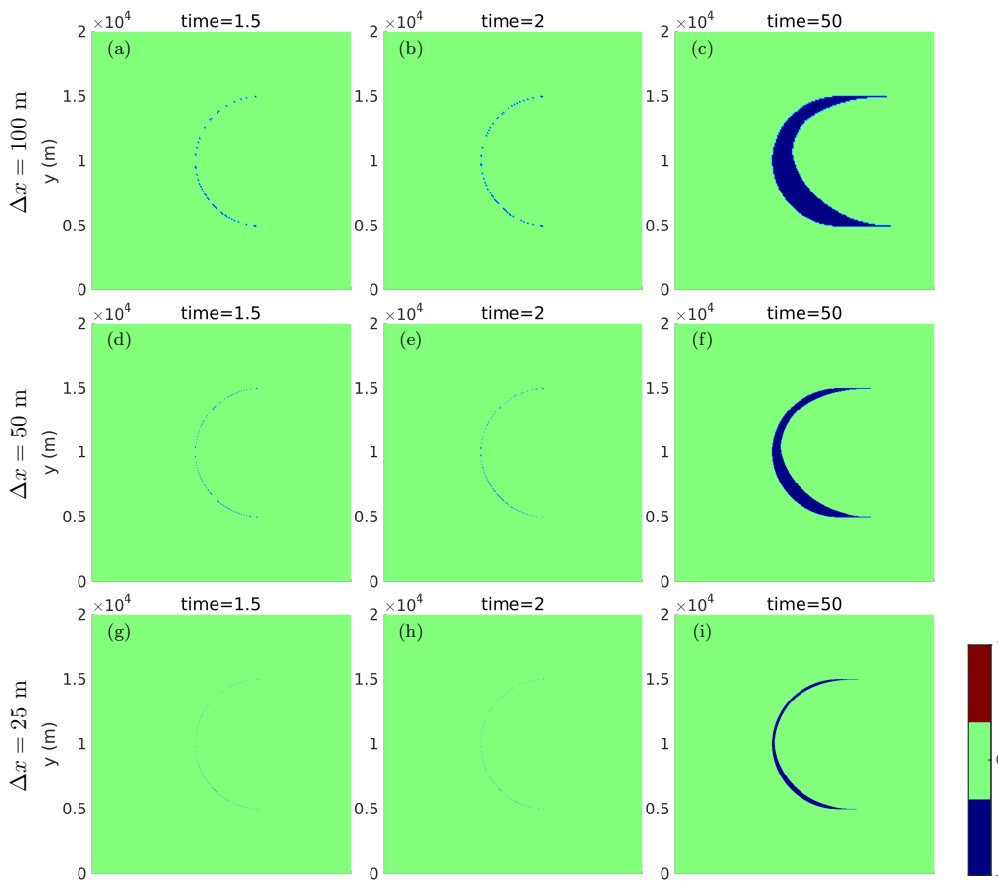

**Figure D4.** Misfit $d(\phi_1, \phi_2)$ of the numerical solution at time $t$ (as $\phi_1$) and its exact solution (as $\phi_2$) on a diagonally aligned triangular mesh using AD stabilization at the reinitialization interval $n_R = 1$, and $v_0 = 1000$ m/a, with mesh size at (a-c) 100 m, (d-f) 50 m, and (g-i) 25 m.

We set $v_0 = 1000$ m/a with the uniform shape of front velocity and conduct simulations at a mesh resolution of 200 m over a 50-year period. Figure E1 illustrates the evolution of the total absolute misfit and their final values at $T = 50$ a.

This experiment exhibits results consistent with other experiments in our study, wherein the SUPG and SUPG+FAB methods with $n_R > 10$ have the smallest misfit areas among all other methods. In terms of magnitude, as discussed in Section 5.3, the errors are scaled by the mean velocity at the front, calculated as $2000 \int_0^{1/2} \sin(2\pi t)\,\mathrm{d}t = \frac{2000}{\pi} \approx 636.56$ m/a during each advance phase and $-636.56$ m/a for the retreat phase. Consequently, these misfits are approximately $0.64$ times those depicted in Figure C1 (a) with the same stabilization and reinitialization intervals.

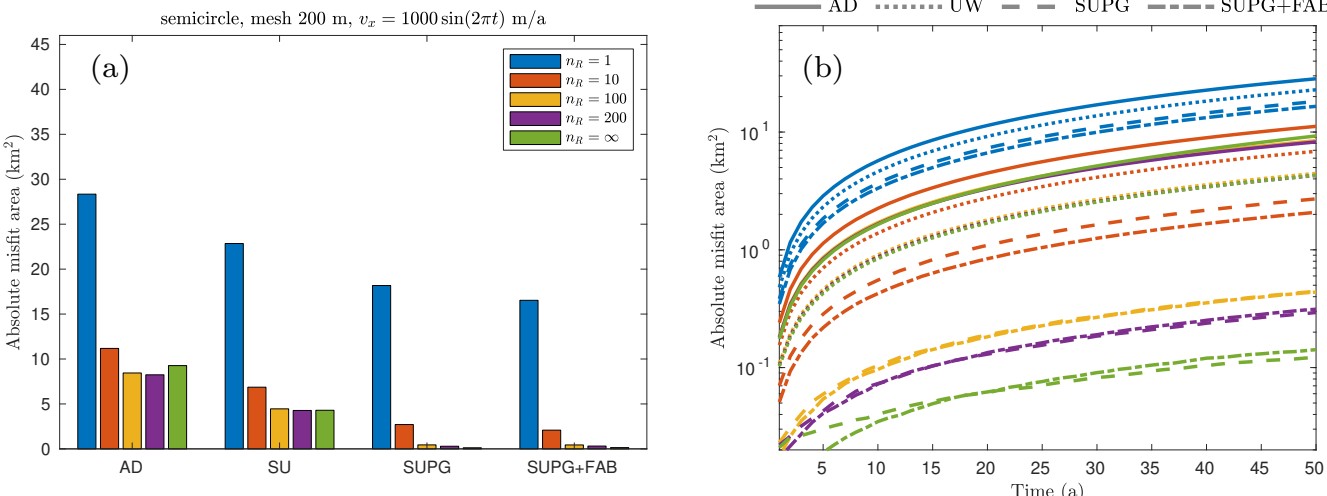

**Figure E1.** (a) The total absolute misfit area at $T = 50$ and (b) the evolution of the total absolute misfit area during the transient simulations with uniform velocity profile at $v(t) = 1000\sin(2\pi t)$ m/a for semi-circle shape ice front at 200 m mesh resolution.

*Author contributions.* GC, MM and HG designed the study. GC did the numerical computations. GC wrote the manuscript with input from MM and HG.

*Competing interests.* The authors declare that they have no conflict of interest.

*Acknowledgements.* This work was supported by the Heising Simons Foundation grant 2019-1161 and 2021-3059. This work is from the PROPHET project, a component of the International Thwaites Glacier Collaboration (ITGC). Support from National Science Foundation (NSF: Grant #1739031) and Natural Environment Research Council (NERC: Grants NE/S006745/1, NE/S006796/1 and NE/T001607/1). ITGC Contribution No. ITGC-XXX

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
