# Peer review of "Numerical stabilization methods for level-set-based ice front migration"

_Geoscientific Model Development, 2023_

## Referee Comment (RC1)

**Review: Numerical stabilization methods for level-set-based ice front migration**

Gong Cheng et al., Geoscientific Model Development, 10.5194/gmd-2023-194

**Matt Trevers (referee)**
matt.trevers@bristol.ac.uk

**General comments**

This study by Cheng et al. concerns itself with investigating the performance of various schemes of numerical stabilization and reinitialization for a level-set method in an ice flow model. Level-set methods are commonly used in ice flow modelling to track the migration of the ice front in response to the ice velocity, and rates of calving and frontal melt. The ice front is defined at the zero contour of the level set function, the motion of which is controlled by an advection equation. This study relates to stabilization and reinitialization procedures applied to the level-set method in two commonly used FEM ice flow models, ISSM and Úa. The authors assess the accuracy of the procedures by applying different combinations of stabilization method and reinitialization interval to an idealized test case with a known solution.

This study is important and novel, and will be a valuable addition to the literature. It has broad application to the field of ice sheet modelling, especially modelling of the outlet glaciers of the Greenland Ice Sheet where ice front migration is a crucial component of the ice flow dynamics. The results of this study demonstrate the importance of the choice of stabilization method and reinitialization interval in minimizing errors.

In general I find this study to be well written and concise. However, I did find some areas where the model description or justification for certain experimental choices wasn't entirely clear, and further detail is required for the sake of clarity. I also identified some questions and areas of interest that I believe could benefit from some further elaboration. Detailed comments are provided below. I am happy to accept this manuscript for publication subject to minor revisions.

**Specific comments**

*Lines 23-26* – There are two sentences here dealing with calving laws and calving rates. The abstract mentioned that the discontinuous nature of calving poses challenges. However this isn't elaborated upon in the main body of the article. Could you include a brief comment here about the implementation of discrete calving laws vs continuous calving rates in models?

*Lines 35-37* – I have a few comments about this sentence. Firstly, it would be better to refer each reference to the stabilization method directly. Secondly, it might be preferable to introduce the acronyms for the stabilization methods later, e.g. in the introductory sentence for Section 2.1, since these three methods plus one extra are the methods applied in this study and the acronyms become the experiment names. You might also consider whether this sentence is a redundant in the introduction and whether it should be replaced with a better introductory sentence for Section 2.1. This comment links to the following comment about the structure of Section 2.1.

*Section 2.1* – The structure of this section needs a bit of reworking for the sake of clarity, to more explicitly state what the four methods applied are. Upon my first readthrough I was left with the impression that only three methods were going to be applied, and only realised my mistake when I got to line 105. In particular, the introductory sentence is very weak. I don't like to see "etc" in a

formal paper. The first sentence should be restructured to explicitly state what the four methods are that will be described in this section, and introduce their acronyms. The descriptions of the methods in the section are generally fine, but care needs to be taken to make it clear that SUPG and SUPG+FAB are distinct methods. Finally, could you state more clearly which experiments are carried out using ISSM and which use Úa. This distinction isn't made except that the FAB method is only applied in Úa. When looking at the results later, it isn't clear which results were derived from Úa and which from ISSM. It would be helpful to include a brief note explaining why the comparison of results derived from two different models is still valid. It may be helpful to include a summary table for this section, but it isn't necessary.

*Section 3* – This section could benefit from some more detail on the experimental design. In particular, could you define the bedrock and ice geometry? I understand that given the prescribed velocities these aren't as crucial as in e.g. a MISMIP-style design, but it's not clear from the description which part of the domain is initially ice-filled and which isn't.

*Line 128* – What is the justification of applying three distinct velocity profiles? The uniform profile should preserve the front shape during advance and retreat while the other two will warp it. Is this the reasoning? If so, why not just two?

*Lines 134-135* – Similarly, why apply two different velocity constants? Is there an *a priori* expectation that the errors will scale linearly with velocity?

*Line 143* – Is there a benefit to fully reversing the velocity field to mimic advance and retreat, as opposed to having a constant flow direction and applying a time-varying calving rate to achieve the same end?

*Section 4* – As mentioned previously, it's not clear which results were produced using Úa or ISSM. However, I think this is best remedied with a change in Section 2.1.

*Figure 2* – Consider a minor rewording to the caption to say "numerical solution".

*Figure 3* – Same as for Figure 2.

*Lines 149-150* – For $n_R$ = 1 the error is visibly non-symmetric in $y$, which isn't the case for $n_R$ = 100. Is there any significance to this?

*Line 164* – There is also visibly less sensitivity to $n_R$ for $v_0$ = 5000 m/a c.f. $v_0$ = 1000 m/a.

*Section 4* – I would be really interested to see somewhere in this section timeseries plots of the evolution of evolution of the total absolute misfit area, either for all the experiments or a selection of them. Does the error increase linearly or exponentially throughout the runs? Does it increase smoothly or do we get abrupt increases associated with the reinitialization interval or the annual cycle?

*Lines 183-185* – Does this explain why there is less sensitivity to $n_R$ for the high-velocity scenario? (See my comment re: Line 164)

*Lines 206-209* – In the previous paragraph, it's mentioned that the errors scale proportionally with mesh spacing for AD and SU. Could you add an equivalent statement to this paragraph about the mesh spacing dependency of the errors in SUPG, for a more direction comparison against AD and SU?

*Section 5.3* – This section seems a bit vague in its conclusions. Is it the form of the velocity profile that matters, or is it just the mean frontal velocity? If the different velocity shapes defined in Table 1

were scaled such that the mean velocities were the same, would we expect differences in the errors to vanish? Given the similarity in results, I'm not convinced that this comparison really enhances our understanding in any meaningful way. If the authors don't wish to completely remove this comparison, it could be simplified by comparing just two velocity profiles rather than three. However, I'm happy to leave this choice to the discretion of the authors.

**Additional comments**

The following comments refer to some questions that occurred to me while reading the manuscript which relate to possible extensions of the study. While these could be answered by carrying out additional experiments, I don't expect the authors to carry out those experiments, and my response to revisions isn't contingent on any additional experiments being run. As such I leave it to the author's discretion how to respond to these questions.

In *Line 164* it is mentioned that all stabilization methods overestimated the ice front advance. If instead the velocity time-cycle were reversed such that the negative velocity is applied first, would we expect to see overestimated retreat rather than advance?

The dependency on mesh spacing is discussed in *Section 5.2*. Were experiments with varying mesh spacing carried out? It would be interesting to see how the errors in the different schemes scale in response to the mesh spacing.

The test case was constructed with simple flow in the one dimension only, and no along-flow gradients. Do the authors think that their conclusions would translate directly to the more complex flow fields in realistic scenarios? Should we expect to see similar relative errors between the different stabilization methods in more realistic scenarios?

**Technical corrections**

*Line 50* – Correct "Method" to "Methods"

*Line 64* – Please reference equations as "Eq. (3)" (mid-sentence) or "Equation (3)" (beginning of sentence). There are numerous other examples of this throughout the manuscript on lines 65, 68, 76, 77, 82, 85, 96, 147, 153, 181, 196, 204, 206 and 207. Please correct these and any other I may have missed.

*Line 68* – Acronyms have previously been defined. See previous comments on Section 2.1.

*Line 76-77* – This sentence is awkward with too many clauses. Please revise for readability.

*Line 92* – "For even values of $p$" reads better at the start of this sentence.

*Line 93* – The FAB acronym was already defined previously.

*Line 104* – "we will" reads better than "we are going to".

*Lines 119 & 120* – Capitalize "Section". Please do the same for any other examples of this that I may have missed.

*Lines 149, 150 & 215* – Please refer to "Figures" when there are multiple. Please do the same for any other examples of this that I may have missed.

*Line 216* – Insert "and" before "both".

*Line 220* – It would be better to start this sentence with "However" instead of "Although".

*Line 234* – Remove repetition of "with".

*Line 233-234* – This sentence needs a bit of revision for readability

---

## Author Comment (AC3)

**Response to the reviewer 1**

November 28, 2023

**General comments**

This study by Cheng et al. concerns itself with investigating the performance of various schemes of numerical stabilization and reinitialization for a level-set method in an ice flow model. Level-set methods are commonly used in ice flow modelling to track the migration of the ice front in response to the ice velocity, and rates of calving and frontal melt. The ice front is defined at the zero contour of the level set function, the motion of which is controlled by an advection equation. This study relates to stabilization and reinitialization procedures applied to the level-set method in two commonly used FEM ice flow models, ISSM and Úa. The authors assess the accuracy of the procedures by applying different combinations of stabilization method and reinitialization interval to an idealized test case with a known solution.

This study is important and novel and will be a valuable addition to the literature. It has broad application to the field of ice sheet modelling, especially modelling of the outlet glaciers of the Greenland Ice Sheet where ice front migration is a crucial component of the ice flow dynamics. The results of this study demonstrate the importance of the choice of stabilization method and reinitialization interval in minimizing errors.

In general I find this study to be well written and concise. However, I did find some areas where the model description or justification for certain experimental choices wasn't entirely clear, and further detail is required for the sake of clarity. I also identified some questions and areas of interest that I believe could benefit from some further elaboration. Detailed comments are provided below. I am happy to accept this manuscript for publication subject to minor revisions.

**Response:** We would like to thank the reviewer for his thoughtful review and positive feedback on our manuscript. The comments and suggestions are addressed below.

**Specific comments**

- Lines 23-26 – There are two sentences here dealing with calving laws and calving rates. The abstract mentioned that the discontinuous nature of

calving poses challenges. However this isn't elaborated upon in the main body of the article. Could you include a brief comment here about the implementation of discrete calving laws vs continuous calving rates in models?

**Response:** Agreed.

- Lines 35-37 – I have a few comments about this sentence. Firstly, it would be better to refer each reference to the stabilization method directly. Secondly, it might be preferable to introduce the acronyms for the stabilization methods later, e.g. in the introductory sentence for Section 2.1, since these three methods plus one extra are the methods applied in this study and the acronyms become the experiment names. You might also consider whether this sentence is a redundant in the introduction and whether it should be replaced with a better introductory sentence for Section 2.1. This comment links to the following comment about the structure of Section 2.1.

  **Response:** We will rephrase this sentence and the introductory sentence in Section 2.1.

- Section 2.1 – The structure of this section needs a bit of reworking for the sake of clarity, to more explicitly state what the four methods applied are. Upon my first readthrough I was left with the impression that only three methods were going to be applied, and only realised my mistake when I got to line 105. In particular, the introductory sentence is very weak. I don't like to see "etc" in a formal paper. The first sentence should be restructured to explicitly state what the four methods are that will be described in this section, and introduce their acronyms. The descriptions of the methods in the section are generally fine, but care needs to be taken to make it clear that SUPG and SUPG+FAB are distinct methods. Finally, could you state more clearly which experiments are carried out using ISSM and which use Úa. This distinction isn't made except that the FAB method is only applied in Úa. When looking at the results later, it isn't clear which results were derived from Úa and which from ISSM. It would be helpful to include a brief note explaining why the comparison of results derived from two different models is still valid. It may be helpful to include a summary table for this section, but it isn't necessary.

  **Response:** Thank you for bringing this to our attention. We will improve Section 2.1 accordingly.

- Section 3 – This section could benefit from some more detail on the experimental design. In particular, could you define the bedrock and ice geometry? I understand that given the prescribed velocities these aren't as crucial as in e.g. a MISMIP-style design, but it's not clear from the description which part of the domain is initially ice-filled and which isn't.

  **Response:** Thank you for pointing this out. We will fill in colors in Figure 1 for the ice-covered region to make it more clear.

- Line 128 – What is the justification of applying three distinct velocity profiles? The uniform profile should preserve the front shape during advance and retreat while the other two will warp it. Is this the reasoning? If so, why not just two?
  **Response:** The three velocity profiles represent zeroth, first, and second order polynomials shape of velocities. We are trying here to capture different ways a terminus may advance and retreat in practice. We will add one sentence of the justification, and reorder Table 1.

- Lines 134-135 – Similarly, why apply two different velocity constants? Is there an a priori expectation that the errors will scale linearly with velocity?
  **Response:** The inclusion of two distinct velocity constants serves to accommodate two distinct scenarios to make sure our conclusions on best practices are robust over a range of possible terminus behavior. There were no specific expectations.

- Line 143 – Is there a benefit to fully reversing the velocity field to mimic advance and retreat, as opposed to having a constant flow direction and applying a time-varying calving rate to achieve the same end?
  **Response:** As explained in Equation (2), the front velocity $\boldsymbol{v}_f$ is the difference between the ice flow velocity and calving rate, therefore, reversing the velocity field $\boldsymbol{v}_f$ is actually equivalent to keep a constant flow velocity and apply a larger calving rate. The reason why we did not want to impose a calving rate on a freely evolving ice sheet model is that it makes it virtualy impossible to find exact solution for ice front motion and so we could not quantify the performance of the different level set schemes.

- Section 4 – As mentioned previously, it's not clear which results were produced using Úa or ISSM. However, I think this is best remedied with a change in Section 2.1.
  **Response:** Agreed.

- Figure 2 – Consider a minor rewording to the caption to say "numerical solution".
  **Response:** Agreed.

- Figure 3 – Same as for Figure 2.
  **Response:** Agreed.

- Lines 149-150 – For $nR = 1$ the error is visibly non-symmetric in y, which isn't the case for $nR = 100$. Is there any significance to this?
  **Response:** We appreciate your observation. The asymmetry observed in $nR = 1$ can be traced back to a mesh effect arising from the structured triangular mesh, where all triangles align diagonally from the top-left to the bottom-right. This effect diminishes with larger reinitialization intervals since the mesh effect is negligible. We will run simulations on an

unstructure mesh and provide additional insights into this phenomenon in the discussion section.

- Line 164 – There is also visibly less sensitivity to $nR$ for v0 = 5000 m/a c.f. v0 = 1000 m/a.
  **Response:** Thank you for pointing this out. We will add a sentence to the result.

- Section 4 – I would be really interested to see somewhere in this section timeseries plots of the evolution of the total absolute misfit area, either for all the experiments or a selection of them. Does the error increase linearly or exponentially throughout the runs? Does it increase smoothly or do we get abrupt increases associated with the reinitialization interval or the annual cycle?
  **Response:** Thank you for mentioning this. The errors increase linearly. We will add the time series plots in the appendix.

- Lines 183-185 – Does this explain why there is less sensitivity to $nR$ for the high-velocity scenario? (See my comment re: Line 164)
  **Response:** Yes, the errors are from different sources.

- Lines 206-209 – In the previous paragraph, it's mentioned that the errors scale proportionally with mesh spacing for AD and SU. Could you add an equivalent statement to this paragraph about the mesh spacing dependency of the errors in SUPG, for a more direct comparison against AD and SU?
  **Response:** As we explained in the previous paragraph, AD and SU modify the weak form of the problem by adding an additional term, which is the dominating source of the error. This additional term scales by the mesh size. Therefore, the numerical errors in these two cases are scaled by the mesh size. However, SUPG modifies the test function, so that the numerical solution actually satisfies the original weak form. Therefore, we would not expect major numerical errors from the stabilization.

- Section 5.3 – This section seems a bit vague in its conclusions. Is it the form of the velocity profile that matters, or is it just the mean frontal velocity? If the different velocity shapes defined in Table 1 were scaled such that the mean velocities were the same, would we expect differences in the errors to vanish? Given the similarity in results, I'm not convinced that this comparison really enhances our understanding in any meaningful way. If the authors don't wish to completely remove this comparison, it could be simplified by comparing just two velocity profiles rather than three. However, I'm happy to leave this choice to the discretion of the authors.
  **Response:** We appreciate your comment. As we explained earlier, these three profiles represent zeroth, first, and second order polynomials to capture different ice front behaviors. We will add a few more sentences about this.

**Additional comments**

The following comments refer to some questions that occurred to me while reading the manuscript which relate to possible extensions of the study. While these could be answered by carrying out additional experiments, I don't expect the authors to carry out those experiments, and my response to revisions isn't contingent on any additional experiments being run. As such I leave it to the author's discretion how to respond to these questions.

- In Line 164 it is mentioned that all stabilization methods overestimated the ice front advance. If instead the velocity time-cycle were reversed such that the negative velocity is applied first, would we expect to see overestimated retreat rather than advance?
  **Response:** This is an interesting point. However, we do not expect to see an overestimated retreat if we flip the sign of the velocity time-cycle. This overestimate of the ice front is due to the convex shape of the ice front.

- The dependency on mesh spacing is discussed in Section 5.2. Were experiments with varying mesh spacing carried out? It would be interesting to see how the errors in the different schemes scale in response to the mesh spacing.
  **Response:** We will run the experiments on 200 m and 500 m resolutions. We expect the results to be similar, they scale by the mesh sizes.

- The test case was constructed with simple flow in the one dimension only, and no along-flow gradients. Do the authors think that their conclusions would translate directly to the more complex flow fields in realistic scenarios? Should we expect to see similar relative errors between the different stabilization methods in more realistic scenarios?
  **Response:** Thank you for highlighting this. Given that the level set function is primarily influenced by the velocity at the zero-level set contour, the numerical errors associated with varying velocities at each time step are expected to follow the patterns identified in this study. While the overall behavior might differ, the inclusion of two velocity constants and consideration of three different orders of polynomials aim to encompass a broad range of common cases.

**Technical corrections**

- Line 50 – Correct "Method" to "Methods"
  **Response:** Agreed.

- Line 64 – Please reference equations as "Eq. (3)" (mid-sentence) or "Equation (3)" (beginning of sentence). There are numerous other examples of this throughout the manuscript on lines 65, 68, 76, 77, 82, 85, 96, 147, 153, 181, 196, 204, 206 and 207. Please correct these and any other I may

have missed.
**Response:** Agreed.

- Line 68 – Acronyms have previously been defined. See previous comments on Section 2.1.
  **Response:** Agreed.

- Line 76-77 – This sentence is awkward with too many clauses. Please revise for readability.
  **Response:** We will change this.

- Line 92 – "For even values of p" reads better at the start of this sentence.
  **Response:** Agreed.

- Line 93 – The FAB acronym was already defined previously.
  **Response:** Agreed.

- Line 104 – "we will" reads better than "we are going to".
  **Response:** Agreed.

- Lines 119 & 120 – Capitalize "Section". Please do the same for any other examples of this that I may have missed.
  **Response:** Agreed.

- Lines 149, 150 & 215 – Please refer to "Figures" when there are multiple. Please do the same for any other examples of this that I may have missed.
  **Response:** Agreed.

- Line 216 – Insert "and" before "both".
  **Response:** Agreed.

- Line 220 – It would be better to start this sentence with "However" instead of "Although".
  **Response:** Agreed.

- Line 234 – Remove repetition of "with".
  **Response:** Agreed.

- Line 233-234 – This sentence needs a bit of revision for readability
  **Response:** We will improve this.

---

## Author Response (AR1)

**Response to the reviewer 1**

January 3, 2024

**General comments**

This study by Cheng et al. concerns itself with investigating the performance of various schemes of numerical stabilization and reinitialization for a level-set method in an ice flow model. Level-set methods are commonly used in ice flow modelling to track the migration of the ice front in response to the ice velocity, and rates of calving and frontal melt. The ice front is defined at the zero contour of the level set function, the motion of which is controlled by an advection equation. This study relates to stabilization and reinitialization procedures applied to the level-set method in two commonly used FEM ice flow models, ISSM and Úa. The authors assess the accuracy of the procedures by applying different combinations of stabilization method and reinitialization interval to an idealized test case with a known solution.

This study is important and novel and will be a valuable addition to the literature. It has broad application to the field of ice sheet modelling, especially modelling of the outlet glaciers of the Greenland Ice Sheet where ice front migration is a crucial component of the ice flow dynamics. The results of this study demonstrate the importance of the choice of stabilization method and reinitialization interval in minimizing errors.

In general I find this study to be well written and concise. However, I did find some areas where the model description or justification for certain experimental choices wasn't entirely clear, and further detail is required for the sake of clarity. I also identified some questions and areas of interest that I believe could benefit from some further elaboration. Detailed comments are provided below. I am happy to accept this manuscript for publication subject to minor revisions.

**Response:** We would like to thank the reviewer for his thoughtful review and positive feedback on our manuscript. The comments and suggestions are addressed below.

**Specific comments**

- Lines 23-26 – There are two sentences here dealing with calving laws and calving rates. The abstract mentioned that the discontinuous nature of

calving poses challenges. However this isn't elaborated upon in the main body of the article. Could you include a brief comment here about the implementation of discrete calving laws vs continuous calving rates in models?

**Response:** Change has been made.

- Lines 35-37 – I have a few comments about this sentence. Firstly, it would be better to refer each reference to the stabilization method directly. Secondly, it might be preferable to introduce the acronyms for the stabilization methods later, e.g. in the introductory sentence for Section 2.1, since these three methods plus one extra are the methods applied in this study and the acronyms become the experiment names. You might also consider whether this sentence is a redundant in the introduction and whether it should be replaced with a better introductory sentence for Section 2.1. This comment links to the following comment about the structure of Section 2.1.

**Response:** We have rephrased this sentence and the introductory sentence in Section 2.1.

- Section 2.1 – The structure of this section needs a bit of reworking for the sake of clarity, to more explicitly state what the four methods applied are. Upon my first readthrough I was left with the impression that only three methods were going to be applied, and only realised my mistake when I got to line 105. In particular, the introductory sentence is very weak. I don't like to see "etc" in a formal paper. The first sentence should be restructured to explicitly state what the four methods are that will be described in this section, and introduce their acronyms. The descriptions of the methods in the section are generally fine, but care needs to be taken to make it clear that SUPG and SUPG+FAB are distinct methods. Finally, could you state more clearly which experiments are carried out using ISSM and which use Úa. This distinction isn't made except that the FAB method is only applied in Úa. When looking at the results later, it isn't clear which results were derived from Úa and which from ISSM. It would be helpful to include a brief note explaining why the comparison of results derived from two different models is still valid. It may be helpful to include a summary table for this section, but it isn't necessary.

**Response:** Thank you for bringing this to our attention. We improved the introduction in Section 2.1, and made it more clear that all the numerical experiments in this paper are solved using ISSM.

- Section 3 – This section could benefit from some more detail on the experimental design. In particular, could you define the bedrock and ice geometry? I understand that given the prescribed velocities these aren't as crucial as in e.g. a MISMIP-style design, but it's not clear from the description which part of the domain is initially ice-filled and which isn't.

**Response:** Thank you for pointing this out. We filled in the colors in

Figure 1 and Figure A1 for the ice-covered region with light blue and the ice-free region with light red to make it more clear.

- Line 128 – What is the justification of applying three distinct velocity profiles? The uniform profile should preserve the front shape during advance and retreat while the other two will warp it. Is this the reasoning? If so, why not just two?
  **Response:** The three velocity profiles represent zeroth, first, and second order polynomials shape of velocities. We are trying here to capture different ways a terminus may advance and retreat in practice. We have added one sentence of the justification, and reordered Table 1.

- Lines 134-135 – Similarly, why apply two different velocity constants? Is there an a priori expectation that the errors will scale linearly with velocity?
  **Response:** The inclusion of two distinct velocity constants serves to accommodate two distinct scenarios to make sure our conclusions on best practices are robust over a range of possible terminus behavior. There were no specific expectations.

- Line 143 – Is there a benefit to fully reversing the velocity field to mimic advance and retreat, as opposed to having a constant flow direction and applying a time-varying calving rate to achieve the same end?
  **Response:** As explained in Equation (2), the front velocity $v_f$ is the difference between the ice flow velocity and calving rate, therefore, reversing the velocity field $v_f$ is actually equivalent to keep a constant flow velocity and apply a larger calving rate. The reason why we did not want to impose a calving rate on a freely evolving ice sheet model is that it makes it virtually impossible to find the exact solution for ice front motion and so we could not quantify the performance of the different level set schemes.

- Section 4 – As mentioned previously, it's not clear which results were produced using Úa or ISSM. However, I think this is best remedied with a change in Section 2.1.
  **Response:** All the experiments are run in ISSM. We made this more clear in the last paragraph of the introduction section.

- Figure 2 – Consider a minor rewording to the caption to say "numerical solution".
  **Response:** Change has been made.

- Figure 3 – Same as for Figure 2.
  **Response:** Change has been made.

- Lines 149-150 – For $n_R = 1$ the error is visibly non-symmetric in y, which isn't the case for $n_R = 100$. Is there any significance to this?
  **Response:** We appreciate your observation. The asymmetry observed in $n_R = 1$ can be traced back to a mesh effect arising from the structured

triangular mesh, where all triangles align diagonally from the top-left to the bottom-right. We have rerun all the simulations on an unstructured mesh and updated all the results accordingly. Notably, there are no major differences between structured and unstructured mesh, except for the non-symmetric error pattern.

- Line 164 – There is also visibly less sensitivity to $n_R$ for v0 = 5000 m/a c.f. v0 = 1000 m/a.
  **Response:** Thank you for pointing this out. We have added a sentence to the result.

- Section 4 – I would be really interested to see somewhere in this section time-series plots of the evolution of the total absolute misfit area, either for all the experiments or a selection of them. Does the error increase linearly or exponentially throughout the runs? Does it increase smoothly or do we get abrupt increases associated with the reinitialization interval or the annual cycle?
  **Response:** Thank you for mentioning this. The errors increase linearly. We have added the time series plots in the appendix.

- Lines 183-185 – Does this explain why there is less sensitivity to $n_R$ for the high-velocity scenario? (See my comment re: Line 164)
  **Response:** Yes, the errors are from different sources.

- Lines 206-209 – In the previous paragraph, it's mentioned that the errors scale proportionally with mesh spacing for AD and SU. Could you add an equivalent statement to this paragraph about the mesh spacing dependency of the errors in SUPG, for a more direct comparison against AD and SU?
  **Response:** As we explained in the previous paragraph, AD and SU modify the weak form of the problem by adding an additional term, which is the dominating source of the error. This additional term scales by the mesh size. Therefore, the numerical errors in these two cases are scaled by the mesh size. However, SUPG modifies the test function, so that the numerical solution actually satisfies the original weak form. Therefore, we would not expect major numerical errors from the stabilization.

- Section 5.3 – This section seems a bit vague in its conclusions. Is it the form of the velocity profile that matters, or is it just the mean frontal velocity? If the different velocity shapes defined in Table 1 were scaled such that the mean velocities were the same, would we expect differences in the errors to vanish? Given the similarity in results, I'm not convinced that this comparison really enhances our understanding in any meaningful way. If the authors don't wish to completely remove this comparison, it could be simplified by comparing just two velocity profiles rather than three. However, I'm happy to leave this choice to the discretion of the authors.
  **Response:** We appreciate your comment. As we explained earlier, these

three profiles represent zeroth, first, and second-order polynomials to capture different ice front behaviors. We have added a sentence about this.

**Additional comments**

The following comments refer to some questions that occurred to me while reading the manuscript which relate to possible extensions of the study. While these could be answered by carrying out additional experiments, I don't expect the authors to carry out those experiments, and my response to revisions isn't contingent on any additional experiments being run. As such I leave it to the author's discretion how to respond to these questions.

- In Line 164 it is mentioned that all stabilization methods overestimated the ice front advance. If instead the velocity time-cycle were reversed such that the negative velocity is applied first, would we expect to see overestimated retreat rather than advance?
  **Response:** This is an interesting point. However, we do not expect to see an overestimated retreat if we flip the sign of the velocity time-cycle. This overestimate of the ice front is due to the convex shape of the ice front.

- The dependency on mesh spacing is discussed in Section 5.2. Were experiments with varying mesh spacing carried out? It would be interesting to see how the errors in the different schemes scale in response to the mesh spacing.
  **Response:** We have run the experiments on 200 m and 400 m resolutions. The results are very similar to the 100 m cases and the errors are scaled by mesh sizes. We added the results in the appendix.

- The test case was constructed with simple flow in the one dimension only, and no along-flow gradients. Do the authors think that their conclusions would translate directly to the more complex flow fields in realistic scenarios? Should we expect to see similar relative errors between the different stabilization methods in more realistic scenarios?
  **Response:** Thank you for highlighting this. Given that the level set function is primarily influenced by the velocity at the zero-level set contour, the numerical errors associated with varying velocities at each time step are expected to follow the patterns identified in this study. While the overall behavior might differ, the inclusion of two velocity constants and consideration of three different orders of polynomials aim to encompass a broad range of common cases.

**Technical corrections**

- Line 50 – Correct "Method" to "Methods"
  **Response:** Change has been made.

- Line 64 – Please reference equations as "Eq. (3)" (mid-sentence) or "Equation (3)" (beginning of sentence). There are numerous other examples of this throughout the manuscript on lines 65, 68, 76, 77, 82, 85, 96, 147, 153, 181, 196, 204, 206 and 207. Please correct these and any other I may have missed.
  **Response:** Change has been made.

- Line 68 – Acronyms have previously been defined. See previous comments on Section 2.1.
  **Response:** Change has been made.

- Line 76-77 – This sentence is awkward with too many clauses. Please revise for readability.
  **Response:** Change has been made.

- Line 92 – "For even values of p" reads better at the start of this sentence.
  **Response:** Change has been made.

- Line 93 – The FAB acronym was already defined previously.
  **Response:** Change has been made.

- Line 104 – "we will" reads better than "we are going to".
  **Response:** Change has been made.

- Lines 119 & 120 – Capitalize "Section". Please do the same for any other examples of this that I may have missed.
  **Response:** Change has been made.

- Lines 149, 150 & 215 – Please refer to "Figures" when there are multiple. Please do the same for any other examples of this that I may have missed.
  **Response:** Change has been made.

- Line 216 – Insert "and" before "both".
  **Response:** Change has been made.

- Line 220 – It would be better to start this sentence with "However" instead of "Although".
  **Response:** Change has been made.

- Line 234 – Remove repetition of "with".
  **Response:** Change has been made.

- Line 233-234 – This sentence needs a bit of revision for readability
  **Response:** Change has been made.

**Response to the reviewer 2**

January 3, 2024

**Summary and High Level Discussion**

The paper explores different stabilization methods for level-set equations and the impact of reinitialization on the accuracy of the solution.

The methods are demonstrated on an idealized geometry (union of a rectangle and a semidisk) to mimic a fjord with a semi-circular ice front. The front velocity is prescribed. The authors nicely present how the different stabilization approaches and the frequency of the reinitialization affect the accuracy of the position of the level set.

The paper addresses a very important topic in ice-sheet modeling and it is easy to read. However, I have major concerns which prevent me from recommending the paper for publication in its present form.

- My main concern is that despite the title and the presentation of the work, there is little about ice front migration in this manuscript. In fact, the geometry and the prescribed velocity are too simplified to be representative of an ice front migration problem. In addition to the very simplified description of the fjord, the prescribed front velocity is aligned with the fjord axis, which is at odds with the fact that the calving component of the front velocity is typically assumed to be orthogonal to the ice front.
  **Response:** Thank you for your insightful review of our manuscript. We appreciate your concerns about the limited representation of ice front migration and the simplicity of the chosen geometry and prescribed velocity. It is important to clarify that this paper is not specifically about calving; rather, its focus is on the treatment of moving boundaries in ice sheet modeling. The deliberate design of our control experiments aims to isolate errors introduced by the numerical treatment of stabilization and reinitialization of the levelset. As demonstrated in the manuscript, these aspects can significantly impact ice front migration if not carefully chosen. We would like to emphasize that the consideration of calving comes after the numerical method is well-tested, which is the step we are taking here.

  The standalone advection level-set equations have been extensively studied in the literature, and this paper adds little to what is already available. On the contrary, I would have found the paper very valuable if the authors

targeted a more realistic ice sheet problem as well, where the level-set velocity was computed using ice flow equations (e.g., the Shallow shelf Approximation) for the ice velocity and at least one of the calving laws typically used in the literature.

**Response:** To the best of our knowledge, research on level-set stabilization and reinitialization in glaciology is scarce, with existing best practices being largely domain-dependent. Notably, level-set equations in other fields primarily address multiphase problems, which substantially differ from ice flow problems. We would greatly appreciate if the reviewer could provide references if we missed important studies. Additionally, considering the ongoing inter-comparison project, CalvingMIP, which focuses on calving and incorporates more realistic ice front geometry, and has already provided an overview over the calving-front implementations in ice-sheet models. Indeed, only two models were found to be currently using the level-set methods, and those are exactly the two presented in this paper, i.e. ISSM and Úa. As part of the CalvingMIP project, the level-set method has now been implemented in the fEthish ice sheet model.This underscores a discernible interest within the ice-sheet modeling community to gain deeper insights into the implementation of calving in models utilizing this approach, and we are therefore addressing an identified need within the community.

- Another concern I have is that the authors do not explain what reinitialization method they are using, despite the fact that the effect of reinitialization is one of the main topics of the paper. When they introduce the reinitialization they reference two papers they co-authored but I could not find any detail there either. Further, plots in figure 2 show a loss of symmetry, which is likely due to the reinitialization procedure, but the authors do not offer any explanation of why that is happening. I worry that there might be an issue with the reinitialization procedure which would affect the results and possibly the paper conclusions.

  **Response:** We acknowledge your valuable observation regarding the need for a more detailed explanation of the reinitialization method, and we provide a thorough description of the method in the revised manuscript. Furthermore, it is important to note that the observed loss of symmetry in Figure 2 can be attributed to a mesh effect stemming from the structured triangular mesh, where all the triangles align diagonally from the top-left to the bottom-right. This effect diminishes when larger reinitialization intervals are employed. We have run additional experiments on 200 m and 400 m resolutions and address these concerns comprehensively in the revised manuscript to bolster the robustness and clarity of our findings.

- Finally, the forward and backward diffusion stabilization considered in this paper aims at keeping the level-set function close to the distance function, so that no reinitialization is needed. This is qualitatively confirmed by their results. However, the authors miss this point in the discussion of the results. Also, the authors do not provide any reference for this stabilization

method.

**Response:** Thank you for bringing this to our attention. The detailed derivations and formulations for the discussed aspects can be found in the Úa Compendium (`https://github.com/GHilmarG/UaSource/blob/master/UaCompendium.pdf`). In the revised manuscript, we have included the appropriate references to enhance the transparency and traceability of our work.

---

## Author Response (AR2)

Dear Editor and Reviewer,

Thank you for your valuable comments. We have improved our manuscript according to the comments and concerns from both of you:

*"1. Address Oversimplification: Revisit the model to ensure it better represents realistic scenarios in ice front migration, testing maybe for another geometry and addressing the oversimplifies velocity field."*

- We would like to highlight that all the parameters chosen in the manuscript (e.g. glacier size, ice front periodicity, ice speed, etc), are representative of Greenland glaciers. We include an additional experiment in the Appendix E to improve the realism of the frontal velocity representation. In this experiment, we introduce seasonal variations by changing the frontal velocity to $v(t)=v_0\sin(2\pi t)$, simulating the dynamic nature of ice front movement influenced by seasonal changes. Notably, the results of this experiment align precisely with those of all other experiments presented in this paper.

2 *"Add a calving term to the velocity profiles, as it is crucial for the realism of the model. Or at least make some systematic investigation on the front dynamic for similar settings."*

- We acknowledge the importance of calving in modeling realistic ice dynamics. However, it is essential to note that the frontal velocity ($v_f = v - c$) used in the level-set equation **implicitly incorporates the effects of calving or calving rate**. Our study primarily focuses on comparing different stabilization and reinitialization strategies for solving the level-set equation. Incorporating a "realistic" calving term may not necessarily provide additional insights into our study, as it would already be accounted for through $v_f$ in the level-set equation. Moreover, introducing a calving law would preclude the availability of analytical solutions, complicating the interpretability of our results. We add some justifications in the discussion section 5.3.

*"3. Provide a more detailed explanation of the level-set function reinitialisation process, possibly exploring different approaches mentioned in literature."*

- Our reinitialization approach is very similar to the one described in \cite{Touré.2016}. We add one paragraph in the method section 2.2 to describe in more detail how we implement the geometric reinitialization algorithm.

*"4. Include a discussion on how the penalization in the stabilization technique might reduce or eliminate the need for reinitialisation."*

- The whole idea of the penalization in the stabilization is to enforce $|\nabla\phi|=1$ (Eikonal equation) when solving the level-set , which is similar to \cite{Hartmann.2010}. We add a paragraph in the discussion section 5.1 to discuss the details.

*"5. Perform additional experiments to show how asymmetry in results reduces with the refinement of the structured mesh, or figure out why there is an asymmetry."*

- The observed asymmetry is attributed to the use of the diagonally aligned triangular mesh. To elucidate this aspect, we incorporate additional experiments in Appendix D featuring a symmetric structured mesh alongside our previously utilized diagonally aligned structured mesh. These supplementary experiments are designed to underscore that the observed asymmetry arises from the specific structure of the diagonally aligned mesh.

*"6. Review and integrate the provided level-set literature into the study to enhance the methodological approach and discussion."*

- We add literature review in the introduction and in the discussion section 5.1

---

## Author Response (AR3)

Dear Dr. Räss,

Thank you for your detailed feedback and for providing your interpretation of the reviewer's comments. We appreciate the time and effort taken to review our revised manuscript and are pleased to hear that it is in much better shape. We are committed to addressing the remaining minor points you have outlined.

1. I recon that it would be valuable to have an experiment to assess whether the asymmetry vanishes on the triangular (diagonally aligned) mesh, upon mesh refinement.
   **Response:** Thank you for the suggestion. We have added an additional experiment in Appendix D, Figure D4, by refining the mesh to 50 m and 25 m. Our results indicate that the asymmetry does not vanish on the diagonally aligned mesh, but it becomes less pronounced as the error decreases with mesh refinement.

2. If relevant, please cite Li et al. with respect to the FAB method.
   **Response:** We added the citation of Li et al. concerning the FAB method.

3. Calving; calving velocity is typically orthogonal to the front, which is at odds with the front velocity being aligned with x. This is a recurring remark and I would also like to see this point addressed. Keeping the benchmark's idealised configuration, it may be relevant to consider some calving orthogonal to the front in order to cover this.
   **Response:** We understand the reviewer's frustration, maybe we have not been clear enough yet. The velocity of the front is *not* the calving velocity. It is a sum of ice speed (which is not necessarily normal to the ice front) and the calving rate (which is generally defined along the normal). So the ice front velocity is not necessarily orthogonal to the front in practice. Note that we did add a case where the velocity of the ice front is orthogonal to the terminus in Appendix A already, so we believe that this case is covered in the manuscript as it stands.
   The main purpose of this manuscript is to show that even with such a 'simple' prescribed frontal velocity, stabilization and reinitialization can a significan impact depending on the choices that are made. We need to make sure the error introduced by the numerical method is under control before moving to more complex situations. As we mentioned, in the CalvingMIP project, we are testing more realistic calving velocity on more complex geometries.

We appreciate your guidance on these points and will address them carefully in our revised manuscript. Thank you for your patience throughout this process. We aim to resubmit the revised manuscript promptly.

Best regards,
Cheng Gong, Mathieu Morlighem, Hilmar Gudmundsson

---

## Author Response (AR4)

Dear Dr. Räss,

Thank you for your prompt review of our revised manuscript and for your constructive feedback.

We appreciate your suggestion to include a brief discussion in the manuscript regarding our response to the third point. We have incorporated a discussion that highlights the distinction between the frontal velocity and calving velocity, as well as our approach in this study. Additionally, we also mentioned the ongoing CalvingMIP project, where we are testing more realistic calving velocities on complex geometries, including both constant and time-dependent calving rates.

Thank you for your continued support and guidance.

Best regards,
Cheng Gong, Mathieu Morlighem, Hilmar Gudmundsson